# A hybrid power load forecasting model using BiStacking and TCN-GRU

Jun Ma *, Jishen Peng, Haotong Han, Liye Song, Hao Liu

Faculty of Electrical and Control Engineering, Liaoning Technical University, Huludao, Liaoning, China

* thisismajunsms@163.com

## Abstract

Accurate power load forecasting helps reduce energy waste and improve grid stability. This paper proposes a hybrid forecasting model, BiStacking+TCN-GRU, which leverages both ensemble learning and deep learning techniques. The model first applies the Pearson correlation coefficient (PCC) to select features highly correlated with the power load. Then, BiStacking is used for preliminary predictions, followed by a temporal convolutional network (TCN) enhanced by a gated recurrent unit (GRU) to produce the final predictions. The experimental validation based on Panama's 2020 electricity load data demonstrated the effectiveness of the model, with the model achieving an RMSE of 29.1213 and an MAE of 22.5206, respectively, with an $R^2$ of 0.9719. These results highlight the model's superior performance in short-term load forecasting, demonstrating its strong practical applicability and theoretical contributions.

## 1 Introduction

Electricity is a vital energy source for modern society and industrial production [1]. Power load forecasting is an essential aspect of optimal electric energy dispatching. It enhances dispatch efficiency, optimizes resource allocation, reduces operating costs, and relieves the pressure on the power grid. As a key component of the national grid, the regional grid is the largest system serving diverse electricity consumers, including residential communities, factories, and central business districts (CBDs) [2]. As a prominent application of time-series forecasting in the power sector [3,4]. The implementation methods for electricity load forecasting are mainly divided into three categories: classical time series forecasting methods, machine learning methods, and deep learning methods [5]. Classical methods, such as autoregressive integrated moving average (ARIMA) and seasonal autoregressive integrated moving average (SARIMA) [6], have proven effective in power load forecasting. Chakhchoukh et al. [7] improved the predictive accuracy of the SARIMA model by filtering outliers in the data. Similarly, Yu et al. [8] enhanced the forecasting performance by integrating the ARIMA model with an artificial neural network (ANN) to better capture seasonal fluctuations, achieving superior accuracy compared to the standalone ARIMA and SARIMA models. Despite these advancements, classical time-series forecasting methods remain limited in addressing the nonlinearities of load curves and fall short of meeting the demands of increasingly complex power grids.

**Data availability statement:** These data were derived from the following resources available in the public domain: Georgia Dataset:(https://transparency.entsoe.eu/load-domain/r2/totalLoadR2/show?name=&defaultValue=false&viewType=TABLE&areaType=B-ZN&atch=false&dateTime.dateTime=01.01.2024+00:00|CET|DAY&bidingZone.values=CTY|10Y1001A1001B012!B-ZN|10Y1001A1001B012&dateTime.timezone=CET_CEST&dateTime.timezone_input=CET+(UTC+1)+/+CEST+(UTC+2) ) Panama Dataset: (https://www.kaggle.com/datasets/saurabhshahane/electricity-load-forecasting)

**Funding:** The author(s) received no specific funding for this work.

**Competing interests:** The authors have declared that no competing interests exist.

Machine learning techniques can utilize a larger number of variables for predictions and typically require less computational time than classical methods when processing large data-sets. Classical machine learning approaches include methods such as random forest (RF) [9] and extreme gradient boosting (XGBoost) [10]. Dudek [11] applied RF for short-term electricity load forecasting, demonstrating its advantages in predicting seasonal cyclic load data. Zhang Rui et al. [12] developed a load forecasting model combining three k-nearest neighbor (KNN) models with different neighboring rules, which were aggregated for final predictions through weighted adjustments. Their results indicated that prediction accuracy was maintained even with limited temperature data. Chen Yongbao et al. [13] enhanced the predictive power of support vector regression (SVR) by integrating the historical temperature data, demonstrating the importance of multi-feature datasets in load forecasting. Abbasi et al. [14] pioneered the application of the XGBoost model to load forecasting, revealing its significant performance benefits in such tasks. Liao Xiaoqun et al. [15] proposed an XGBoost-based load forecasting model that incorporated the similar days by analyzing common meteorological patterns and daily types, improving the accuracy for short-term load forecasting. Yang Yang et al. [16] proposed a model that combines an extremely robust extreme learning machine (ELM) with an improved whale optimization algorithm (WOA). This model has achieved excellent performance in load forecasting.

The development of deep learning has provided strong support for electricity load forecasting and has become the primary choice of many researchers today [17], advancements in deep learning have significantly enhanced power load forecasting by leveraging models such as recurrent neural networks (RNNs) [18], long short-term memory (LSTM) [19], and temporal convolutional networks (TCNs) [20]. These iterations improve the prediction accuracy, streamline the model structures, and reduce computational complexity. Sun Gaiping et al. [21] employed the feature selection based on maximum correlation and combined LSTM and RNN models for load prediction. Alhussein et al. [22] integrated the convolutional neural networks (CNNs) with LSTM to create a robust deep learning model for power load forecasting. Veeramsetty et al. [23] enhanced forecasting by reducing the input dimensions of the gated recurrent unit model using RF. Wang Yuanyuan et al. [24] extracted the hidden information from power features using TCN and combined it with a lightweight gradient boosting machine for short-term load forecasting, demonstrating the effectiveness of merging deep learning with machine learning to address complex forecasting tasks. Yang Yang et al. [25] proposed a multi-scale deep neural network (MscaleDNN) with an attention mechanism, demonstrating the superior performance of multi-scale deep neural networks in load forecasting tasks.

The evolution of power load forecasting technology is characterized by three major approaches: classical time-series forecasting methods, machine learning algorithms, and deep learning techniques. Each method has distinct strengths and limitations, and relying on a single algorithm often results in suboptimal performance. Combining multiple models can significantly enhance prediction accuracy. Ensemble learning, a key concept in this regard, integrates multiple weak learners to create a strong learner, thereby improving overall predictions. This approach effectively addresses the limitations of individual learners, with basic ensemble models composed of identical base learners outperforming single models through appropriate configuration and training. Common ensemble learning methods include Bagging, Boosting, and Stacking [26], each of which is capable of delivering a higher prediction accuracy. Deng Xuzhi et al. [27] developed a model for predicting the spike load times under extreme weather by combining Bagging and XGBoost. Tan Zhenqi et al. [28] designed an SVR-based model with various kernel functions, using Stacking to merge base models for exceptional forecasting performance. Guo Fusen et al. [29] proposed a Stacking model that integrated ANNs, XGBoost, LSTM, StackedLSTM, and Bi-LSTM as the base learners, improving the accuracy by

incorporating the meteorological factors. Shi Jiaqi et al. [30] proposed a diversity regularized Stacking method that improves electric load forecasting accuracy through tree-based embedded feature selection and diversity regularization. Luo Shucheng et al. [31] proposed a short-term electric load forecasting algorithm based on a CNN-BiLSTM-Attention and XGBoost Stacking ensemble method. This algorithm effectively captures the spatial and temporal dependencies in load data, thereby improving the prediction accuracy and robustness.

The ensemble learning algorithm improves the accuracy and robustness of power load forecasting by integrating the predictions from multiple models. Through multilayer neural networks, deep learning can effectively capture nonlinear relationships in data and excels at handling complex datasets. This study proposed a hybrid approach that combines ensemble learning with deep learning, leveraging ensemble models to reduce data complexity and computational costs while enhancing the performance of power load forecasting models.

Regional differences can be influenced by certain climatic factors, such as latitude, altitude, rainfall, and wind, which significantly affect power loads in load forecasting studies [32]. Integrating climate data into the forecasting process facilitates predictions that reflect actual conditions more accurately, thereby enhancing the accuracy. This study employed the PCC for feature engineering to select relevant features and improve the predictive capability.

This study introduced the BiStacking+TCN-GRU hybrid model by integrating an ensemble learning model with a deep learning model. The BiStacking component utilized RF, KNN, and XGBoost as the base learners within a two-layer structure, pairing each base learner one-to-one and selecting the optimal combinations based on the prediction results. The BiStacking output was then fed into the TCN-GRU model to generate the final predictions. This approach not only improved prediction accuracy but also reduced computational resource consumption and enhanced operational efficiency. In practical applications, the model enabled precise power load forecasting and optimized power production and distribution to improve the reliability of grid operations.

In summary, the main contributions of this study are as follows:

(1) The application of feature engineering markedly improved the prediction accuracy of the model, demonstrating its substantial positive impact on forecasting outcomes.

(2) The BiStacking model proposed in this study differed from traditional stacking models by utilizing a two-layer base learner structure, which significantly enhanced the prediction accuracy of the ensemble learning model.

(3) The integration of the TCN and GRU effectively preserved the distant contextual information in the time-series data with extended time spans, thereby enhancing the model's capability to capture long-term dependencies.

(4) Robustness experiments demonstrated that the proposed model possessed strong generalization capabilities across various datasets.

The remainder of this paper is organized as follows. The section on Prework of Forecasting details the principles underlying the algorithms used in feature engineering and explains the fundamental principles and methods of the proposed model. The section on Fusion short term load forecasting model construction outlines the model construction process, including the functions of its components and the complete prediction workflow. The section on Experimental analysis presents control experiments to evaluate the performance of the model, assesses its predictive effectiveness through data analysis, and conducts robustness experiments to verify its generalization capabilities. Finally, the section on Conclusion and Future Research Directions concludes the study and explores potential future research directions.

## 2 Prework of forecasting

This section describes the feature engineering processes for raw data, along with the TCN and Stacking ensemble learning methods, which together form the prediction model proposed in this study.

### 2.1 Data analysis and feature engineering

In time-series prediction tasks, the quality of data has a significant impact on forecast accuracy. High-quality data directly influences the upper bound of prediction accuracy. Effective data analysis and feature engineering are essential for extracting features that are highly relevant to the research objectives, as well as for eliminating noise and reducing data complexity. These processes enhance both the computational efficiency and the prediction accuracy of the model. Specifically, feature selection helps to reduce the number of parameters the model has to learn, avoiding the learning of irrelevant or redundant information, thereby reducing the risk of overfitting. Moreover, feature selection contributes to reducing computational complexity. This study employed PCC for feature selection by calculating the correlation coefficients of each feature in the original dataset. This method identifies the features that positively influence the prediction outcomes, thereby enhancing the computational efficiency and generalization ability of the model (Equation (1)).

$$r = \frac{\sum(x_i - \bar{x})(y_i - \bar{y})}{\sqrt{\sum(x_i - \bar{x})^2 \sum(y_i - \bar{y})^2}} \tag{1}$$

where $x_i$ and $y_i$ are the data points of the two variables, respectively. $\bar{x}$ and $\bar{y}$ are the mean values of the two variables, respectively. $r$ represents the computed correlation coefficient.

### 2.2 Base model and algorithm

This section introduces the foundational models and algorithms used in this study. It begins with a detailed description of the structure and principles of the TCN model, highlighting its advantages in processing time-series data, particularly in sequence modeling and feature extraction. Next, the basic concept of the Stacking model is presented, along with an explanation of how ensemble learning enhances prediction performance. Through the analysis of these two models, this section provides the theoretical foundation for the modeling methods and load forecasting experiments discussed in subsequent chapters.

**2.2.1 TCN.** TCN is a deep learning-based convolutional neural network that employs one-dimensional convolutional layers to extract local features from time-series data. By stacking multiple convolutional layers and applying nonlinear activation functions, the TCN can effectively extend its receptive field, enabling it to capture information across various time scales (Fig 1).

TCN utilizes dilated convolutions to expand its receptive field, with the convolution operation at time $t$ described by Equation (2).

$$\text{F}(t) = (x * df)(t) = \sum_{i=0}^{k-1} f(i) \cdot x_t - x_{t-d \cdot i} \tag{2}$$

where $x$ is the input sequence, $*$ denotes the convolution operation, $d$ is the dilation factor determined by the number of convolution layers, $k$ is the filter size, $f(i)$ is the $i$-th element of the filter, and $x_{t-d \cdot i}$ is the element of the input sequence corresponding to the filter.

TCN utilizes Residual Blocks to improve the training of deeper networks, thereby enhancing the learning capacity and incorporating Shortcut Connections. Each module comprises

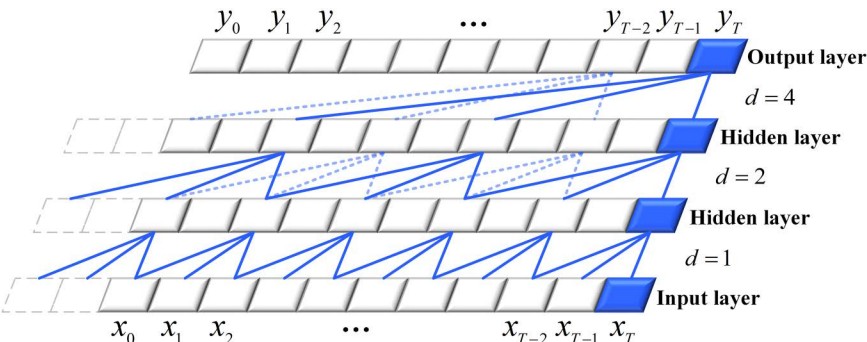

**Fig 1. Dilated Convolutional Structure.**

a causal convolution layer, normalization layer, activation function (typically ReLU), and dropout layer. A residual connection was formed by summing the activation output with the original input. The causal convolution layer ensures that the output depends only on the current and previous input sequences, thus meeting the causality requirement for time-series modeling. In addition, the ReLU activation function can mitigate the vanishing gradient problem during deep network training. The structure of the model is illustrated in Fig 2.

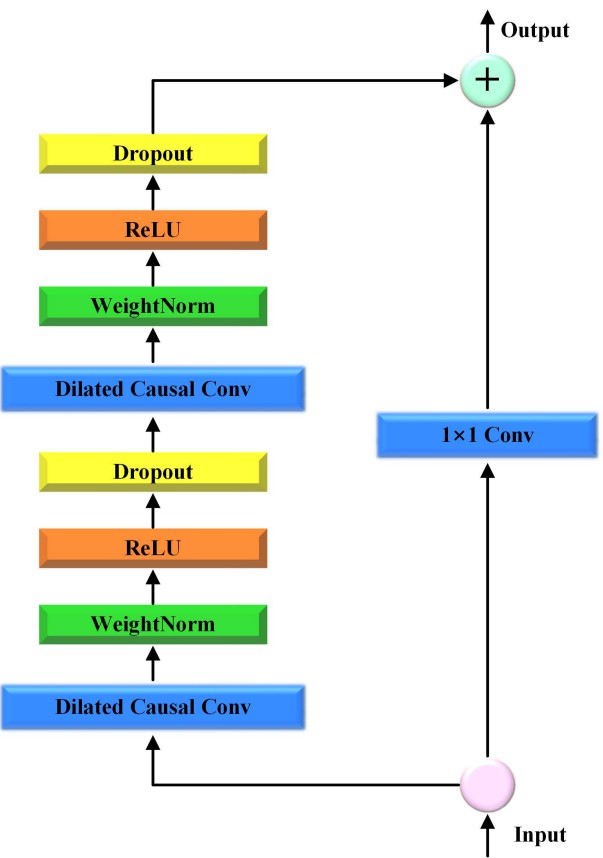

**Fig 2. Structure of TCN.**

**2.2.2 Ensemble learning based on Stacking.** Ensemble learning is based on statistical learning theory, which combines multiple algorithms to improve the prediction accuracy. Bagging and Boosting exhibit several notable limitations compared to Stacking. These methods primarily construct base learners using parallel or sequential approaches, and their prediction accuracy often experiences diminishing returns when additional models are used. In contrast, Stacking enables more effective integration of base learners, resulting in improved predictions, and offers greater flexibility in model selection by accommodating a diverse range of models. The basic Bagging and Boosting models are illustrated in Fig 3.

The Stacking prediction model follows a two-layer framework. The first layer contains multiple base learners that can generate predictions that serve as inputs for the second layer. The second layer, known as the meta-learner, can produce final predictions. This approach optimizes the arrangement of the base learners, thereby improving overall prediction accuracy (Fig 4).

The specific training method for stacking is as follows.

The dataset $S = \{(y_n, x_n), n = 1, \cdots, N\}$, where $x_n$ is the feature vector of the $n$-th sample and $y_n$ is the prediction value corresponding to the $n$-th sample, is randomly partitioned into $K$ subsets of approximately equal size: $S_1, S_2, \cdots, S_k$. In this study, $S_{-k} = S - S_k$, where $S_k$ and $S_{-k}$ denote the $k$-fold test set and the training set, respectively, in k-fold cross-validation. For the layer 1 prediction algorithm containing $K$ base learners, the training set $S_{-k}$ is used to train the $k$-th base learner to obtain the base model $L_k$ where $k = 1, \cdots, K$.

For each sample $x_n$ in the $k$-th fold test set $S_k$ during k-fold cross-validation, the prediction made by the base learner $L_k$ is denoted as $Z_{kn}$. After completing the cross-validation process, the output data of the $K$ base learners are combined to form a new data sample $S_{new} = \{(y_n, z_{1n}, z_{1n}, \cdots, z_{kn}), n = 1, \cdots, N\}$.

$S_{new}$ is the input data for Stacking layer 2, where the meta-learner $L_{new}$ is trained using $S_{new}$. The meta-learner in Layer 2 enhances the prediction accuracy by utilizing the outputs of the base learners in Layer 1 as inputs, enabling it to identify and correct the prediction errors of Layer 1 algorithms.

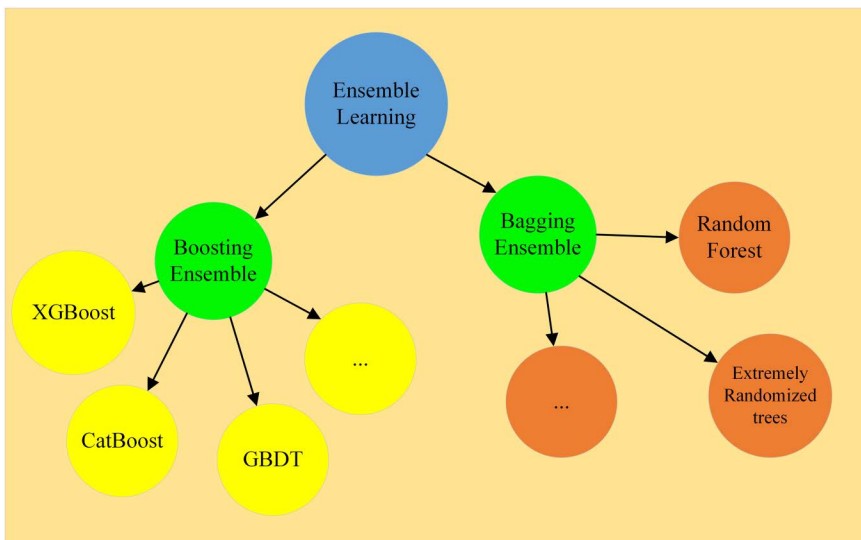

**Fig 3. Classification of Ensemble Learning.**

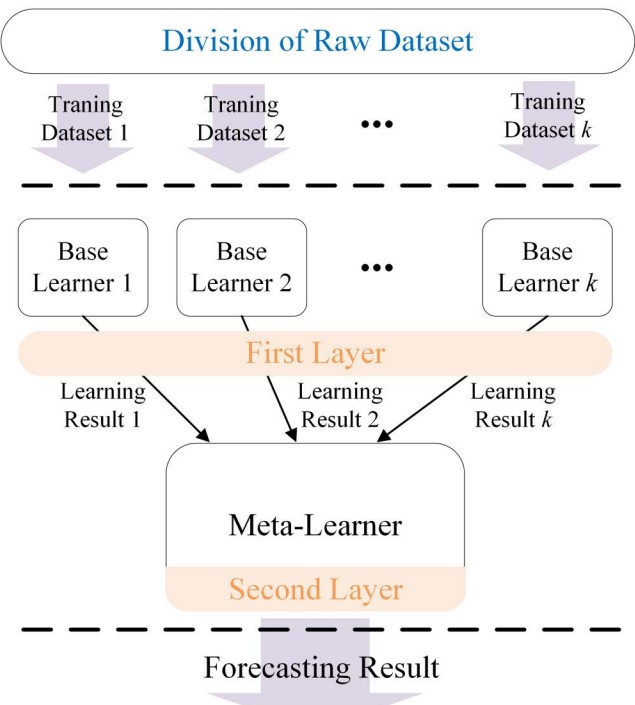

**Fig 4. Basic framework of stacking.**

## 3 Fusion short term load forecasting model construction

This section details the enhancements made to the key components of the model for the forecasting task and outlines the complete process of power load forecasting performed by the model.

### 3.1 Step 1: Stacking part construction

Guo Fusen et al. [29] demonstrated the superiority of the Stacking model in load forecasting. However, the Stacking model and its single-layer base learners have certain shortcomings, particularly when the base learners are highly correlated, which may result in overfitting and exacerbate the bias-variance trade-off. Moreover, due to the relatively simple structure of the single-layer base learners, they have not been able to fully capture high-dimensional features and complex nonlinear relationships, which limits the model's performance in handling complex forecasting tasks. Nevertheless, based on the current model architecture, there is still potential for further improvement and optimization. Therefore, given that the performance of base learners can significantly influence the overall effectiveness of the stacking model, this study replaced the conventional single-layer base learner structure. A Bi-directional Selection Layer was introduced in the first layer of the stacking model, incorporating six high-performance models, including XGBoost and LSTM, as the base learner components.

To achieve the optimal combination of base learners within the Bi-directional Selection Layer, a randomized strategy was employed, pairing two base learners in bi-directional combinations. Using the same dataset for training, 21 combinations were evaluated, and the top six based on prediction accuracy were selected as the base learners for Stacking. Referred to as

Tandem Base Learners owing to their tandem connection characteristics, their structure and construction are illustrated in Fig 5.

The meta-learner for the second layer of the stacking structure was selected based on its strong generalization capabilities. This choice enabled the model to generalize and correct biases across multiple learning algorithms relative to the training set while preventing overfitting through aggregation. Consequently, XGBoost was utilized as the meta-learner for the second layer.

## 3.2  Step 2: Deep learning part construction

Jiarui Tian et al. [33] combined TCN with LSTM to enhance the precise forecasting capability of electric vehicle charging load, demonstrating the advantages of TCN combined with RNN-based models in extracting time-series features. Hu Xiaoyan et al. [34] utilized a combination of TCN and GRU to effectively capture long-term dependencies within time-series data, further improving the accuracy of prediction results. These studies provide valuable experience for the integration of TCN with RNN-based models. Therefore, this paper combines the feature extraction strengths of TCN with the ability of GRU to model long-term dependencies, resulting in the development of the TCN-GRU model.

GRU can effectively capture the long-term dependencies in sequential data and address certain challenges such as gradient vanishing and gradient explosion through its gating mechanisms, including update and reset gates. The detailed structure of the GRU is shown in Fig 6.

In the figure, $\otimes$ denotes the element-wise multiplication of the matrix elements; $\oplus$ denotes the matrix addition operation; and $\sigma$ represents the logistic sigmoid function that transforms the data to a value in the range of 0–1, thus serving as a gating signal.

The hidden state $h_{t-1}$ from the previous unit carries relevant information from the preceding time step. Combined with the current input $x_t$, the GRU calculates the current hidden

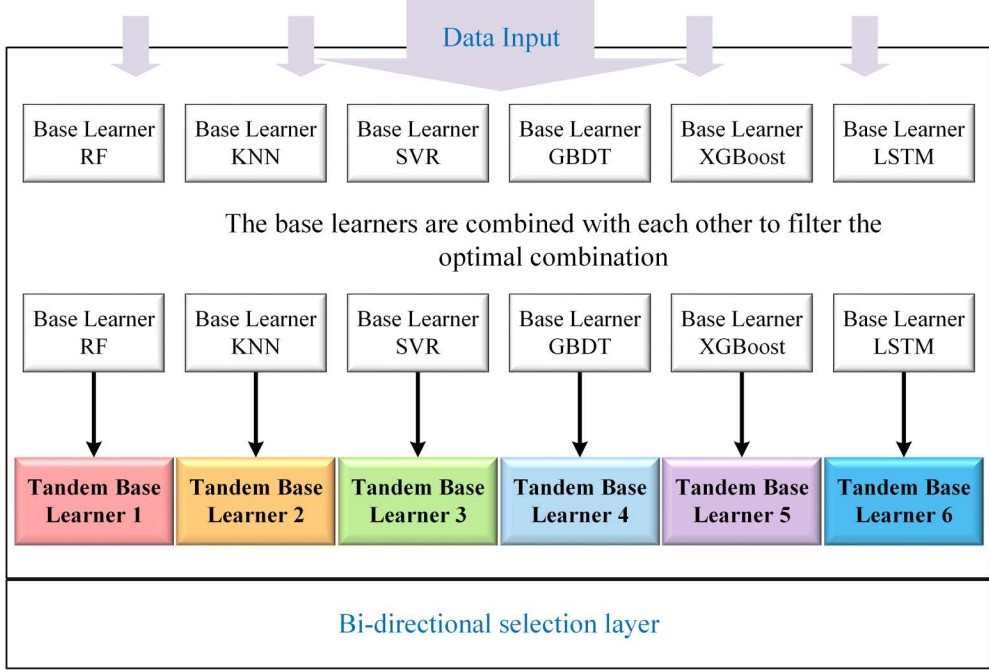

**Fig 5.  Bi-directional selection layer.**

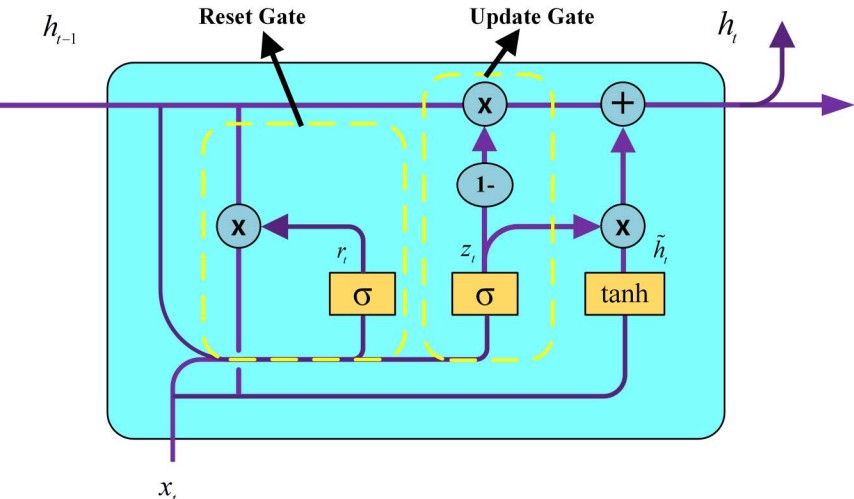

**Fig 6. Structure of GRU.**

state output $y_t$ and the hidden state to be passed to the next unit $h_t$. The gating signals were derived from $h_{t-1}$ and $x_t$. $W_r$ is the weight matrix associated with the input and the hidden state from the previous time step, which is used to compute the reset gate. Therefore, the reset gate computation is formulated as shown in Equation (3).

$$r_t = \sigma(W_r[h_{t-1}, x_t] + b_r) \tag{3}$$

After computing the gating signal $r_t$, the candidate hidden state $\tilde{h}_t$ was determined using Equation (4). $\tilde{h}_t$ primarily reflects the current input data $x_t$, representing the current state. A larger value of $r_t$ signifies a greater retention of information from the previous time step. $W_h$ and $W_z$ are also weight matrices, used for computing the candidate hidden state and the update gate, respectively. Therefore, the update gate computation is formulated as shown in Equation (5).

$$\tilde{h}_t = \tanh(W_h[r_t * h_{t-1}, x_t] + b_h) \tag{4}$$

$$z_t = \sigma(W_z[h_{t-1}, x_t] + b_z) \tag{5}$$

The gating signal $z_t$ determines data retention, with values closer to 1 retaining more data and values closer to 0 discarding more data. GRU leverages this gating mechanism to enable simultaneous memory updating and forgetting. In the above equations, $b_r$, $b_h$, and $b_z$ represent the bias terms for the reset gate, candidate hidden state, and update gate in the GRU, respectively. Along with the weight matrices, they help the model adjust the activation values of each gate during the training process. Following the application of the update gate, the current hidden state is calculated using Equation (6) along with the current candidate hidden state.

$$h_t = (1 - z_t) * h_{t-1} + z_t * \tilde{h}_t \tag{6}$$

where $(1 - z_t) * h_{t-1}$ represents the selective forgetting of the previous hidden state, and $z_t * \tilde{h}_t$ represents the selective incorporation of the current candidate hidden state, which contains information about the current time step.

In the deep learning component of the model, the TCN output serves as the input to the GRU, adopting the GRU's memory capabilities for enhanced sequence modeling and prediction. This architecture is computationally efficient, consumes minimal memory, and effectively captures the long-term dependencies. Its structure is shown in Fig 7.

The hidden layers in the example consisted of two layers. In the dilated convolution operation, the dilation factor $d$ is set to $2^{n-1}$. Assuming that the extracted data for hidden layer 1 span from $h_{11}$ to $h_{1T}$, the figure shows that $h_{11}$ captures the data features of $x_T$, $x_{T-1}$, and $x_{T-2}$. Similarly, in hidden layer 2, $h_{2T}$ captures the data features of $h_{1T}$, $h_{1T-2}$, and $h_{1T-4}$, indicating that the receptive field of hidden layer 2 ranges from $x_T$ to $x_{T-6}$. As the dilation factor increased, the model could generate predictions over a longer time series. The number of neurons in the GRU unit matches and corresponds one-to-one to the final output layer. The output layer redistributes data through the gating unit, assigning higher weights to historical data with a significant predictive impact and lower weights to less impactful data. The GRU-predicted output spans from $y_0$ to $y_T$.

### 3.3 Framework of the forecasting model

Building upon the previous discussion of the TCN-GRU model, this paper proposes a novel forecasting model that integrates the BiStacking model with the TCN-GRU model. This model integrates the ensemble advantages of BiStacking with the feature learning capabilities of deep learning, aiming to improve the model's ability to handle complex data patterns and enhance prediction performance.

This subsection outlines the workflow of the model. First, the raw power load dataset was processed using the Pearson correlation coefficient to identify the most highly correlated features. These selected features were input into the Stacking Part for initial predictions. The outputs from the Stacking Part were then fed into the Deep Learning Part for further processing via TCN-GRU, producing the final prediction. To prevent overfitting, K-fold cross-validation was employed in the Stacking Part. This method dynamically adjusted the validation set by repeatedly splitting the training and validation datasets, reducing reliance on specific features, and mitigating overfitting. Specifically, the training dataset was divided into six subsets. Each Tandem Base Learner sequentially employed one subset as the validation set while training on the remaining subsets. This ensured that each subset served as the validation set in turn, without overlap between validation sets across Tandem Base Learners. The prediction outputs

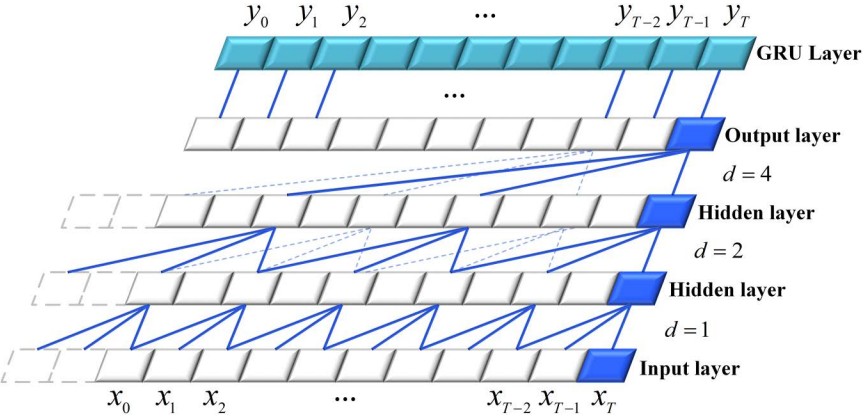

**Fig 7. Structure of TCN-GRU.**

of the Tandem Base Learners for their respective validation datasets were combined to create a new dataset with the same size as the original dataset.

In the Deep Learning Part, input data are normalized to minimize the risk of overfitting by scaling feature values to a uniform range. Fig 8 illustrates the dataset division for K-fold cross-validation and architecture of the proposed model.

## 4 Experimental analysis

This section presents the power load dataset used in the experiments, performance evaluation metrics, parameter settings for each model, experimental results and analyses, and robustness experiments. These elements validate the predictive performance and practical applicability of the proposed model.

### 4.1 Data sources

This study utilized Panama's 2020 hourly electricity load data, which spans a long period and exhibits significant seasonal fluctuations and complex load variation patterns. Additionally, the dataset includes multiple features, making it well-suited for validating the effectiveness of short-term load forecasting models. The data comprises 8,760 data points. The dataset from January to December 2020 was divided into training and test sets with a 7:3 split, allocating 6,100 samples to the training set and 2,660 to the test set. It included features such as weather and humidity, with normalization applied to standardize the feature magnitudes. Details of the dataset composition are presented in Table 1.

### 4.2 Evaluation metrics

To evaluate model performance objectively, this study employed three metrics: R-squared ($R^2$), Mean Absolute Error (MAE), and Root Mean Square Error (RMSE), each offering a unique perspective.

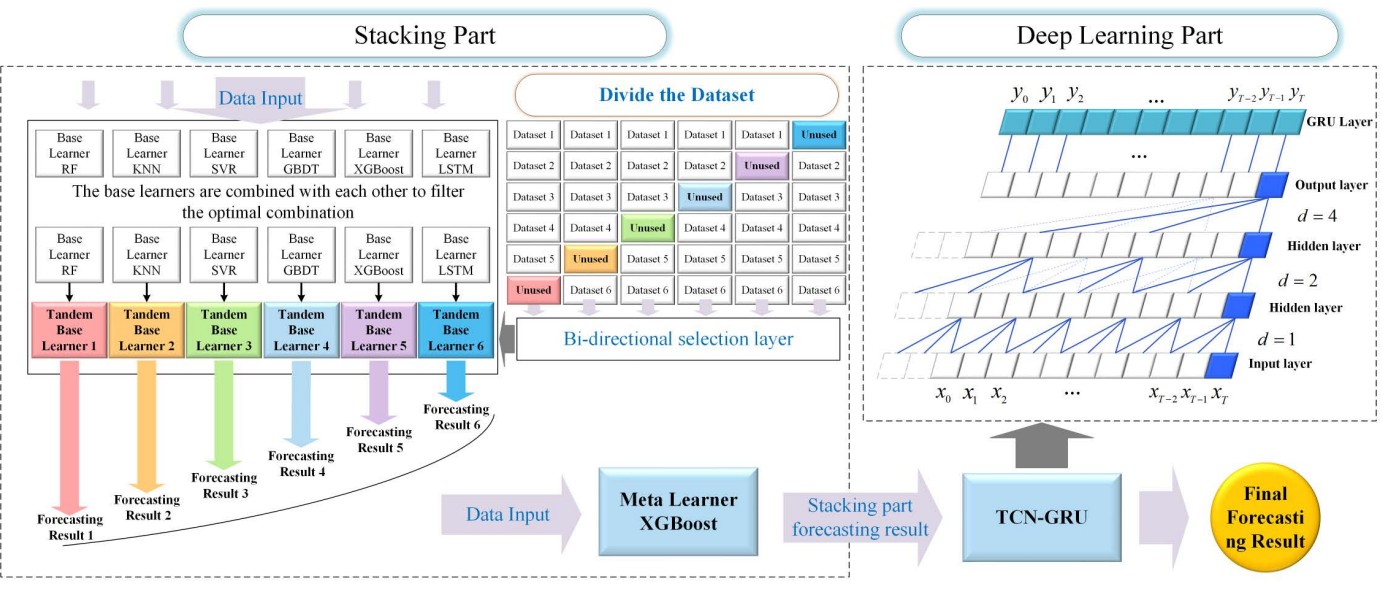

**Fig 8. Research framework for BiStacking +TCN-GRU.**

**Table 1. Statistical results of raw data.**

| Season | Number of Samples | Maximum value | Minimum value | Mean Value | Sampling Frequency |
|---|---|---|---|---|---|
| Spring | 2208 | 1574.42 | 666.7528 | 1132.1951 | 1h |
| Summer | 2208 | 1645.4773 | 674.4174 | 1149.2518 | 1h |
| Fall | 2184 | 1531.2143 | 708.8303 | 1122.0724 | 1h |
| Winters | 2160 | 1540.6727 | 509.8358 | 1135.1960 | 1h |

$R^2$, also known as the coefficient of determination, is a key metric for assessing the goodness of fit of a regression model. Its calculation method is shown in Equation (7), and its value ranges from 0 to 1, where a value closer to 1 indicates a better model fit. $R^2$ provides an intuitive measure of model fit, but it cannot be used alone to evaluate predictive accuracy. A model may achieve a high $R^2$ value, but if the prediction errors are large (e.g., high RMSE and MAE), it may still fail to provide reliable forecasts. Therefore, when assessing a model's predictive performance, $R^2$ should be considered alongside RMSE and MAE.

$$R^2 = 1 - \frac{\sum_{i=1}^{n}(y_i - \hat{y}_i)^2}{\sum_{i=1}^{n}(y_i - \overline{y})^2} \tag{7}$$

MAE measures the difference between the predicted and actual values, with its calculation method shown in Equation (8). MAE is computed as the mean of the absolute values of all prediction errors. MAE is a simple and easy-to-interpret metric since it assigns equal weight to all errors and does not place more emphasis on larger errors. A lower MAE value indicates smaller prediction errors and higher forecasting accuracy. MAE is particularly useful in scenarios where a balanced evaluation of all errors is required and is effective in mitigating the impact of outliers.

$$MAE = \frac{1}{n}\sum_{i=1}^{n}\left|y_i - \hat{y}_i\right| \tag{8}$$

RMSE is a metric used to measure the prediction accuracy of a forecasting model, with its calculation method shown in Equation (9). RMSE is the square root of the mean of the squared prediction errors. A smaller RMSE value indicates a smaller difference between the predicted and actual values, leading to higher prediction accuracy and a better reflection of the model's forecasting quality. Since RMSE effectively balances overall prediction accuracy and extreme errors, it is particularly suitable for tasks that require high precision in forecasting.

$$RMSE = \sqrt{\frac{1}{n}\sum_{i=1}^{n}(y_i - \hat{y}_i)^2} \tag{9}$$

where $y_i$ is the normalized actual value, $\hat{y}_i$ is the normalized predicted value, $\overline{y}$ is the mean value, and $n$ is the number of samples.

### 4.3 Analysis of experimental results

This section experimentally validates the predictive capability and robustness of the proposed model. Specifically, Section 4.3.1 Experimental setup and Section 4.3.2 Validation of feature

engineering introduce the experimental setup and data preprocessing, and verify the effectiveness of feature engineering. Section 4.3.3 Deep learning model selection demonstrates the performance advantages of the chosen combination model by comparing it with various deep learning models. Sections 4.3.4 BiStacking+TCN-GRU experimental results to 4.3.6 Robustness analysis focus on presenting the experimental results of the BiStacking+TCN-GRU model, and further explore the model's stability and generalization ability through error bars and robustness analysis. The experimental results show that the proposed model performs excellently in all tasks, demonstrating strong predictive capability and high reliability, thereby validating its potential for practical applications.

**4.3.1 Experimental setup.** Simulation experiments were conducted to evaluate the feasibility of the proposed model using a computer system with the following configuration: Windows 10, Intel Core i7 CPU, and Python 3.9 as the programming language.

The BiStacking model was implemented using the scikit-learn 1.4.0 library and pandas 1.1.5. It incorporated six base learners: random forest, KNN, GBDT, SVM, XGBoost, and LSTM. The specific parameters of each model are detailed below.

**Random Forest:** The Random Forest model parameters for this test were configured as follows: the number of trees was set to 50, maximum tree depth was 10, minimum number of samples required to split an internal node was 2, and minimum number of samples required at a leaf node was 1.

**KNN:** The KNN model parameters for this test were configured as follows: the number of neighbors was set to 5, the leaf size was 30, and the power parameter for the Minkowski metric was 2.

**GBDT:** The GBDT model parameters for this test were configured as follows: the number of boosting stages was set to 50, learning rate was 0.01, and maximum depth of the individual regression estimators was 10.

**SVR:** The SVR model parameters for this test were configured as follows: the kernel cache size was set to 50, and the hard limit on solver iterations was -1.

**XGBoost:** The XGBoost model parameters for this test were configured as follows: the learning rate was set to 0.01, number of trees was 50, and maximum tree depth was 10.

**LSTM:** The LSTM model parameters for this test were configured as follows: "ReLU" was used as the activation function, with a learning rate of 0.01 and dropout rate of 0.2. The number of training epochs was set to 50, "Adam" was used as the optimizer, the loss function was MSE, and the batch size was 32.

**TCN:** The TCN model parameters for this test were configured as follows: the number of filters was 64, dropout rate was 0.2, kernel size was 3, and number of training epochs was 50. "Adam" was used as the optimizer, the loss function was MSE, and the batch size was 32.

**GRU:** The GRU model parameters for this test were configured as follows: the number of hidden units was set to 10, dropout rate was 0.2, the output layer contained 1 neuron, and number of training epochs was 50. "Adam" was applied as the optimizer, the loss function was MSE, and the batch size was 32.

**4.3.2 Validation of feature engineering.** To evaluate the impact of feature engineering on the prediction accuracy of the model, Pearson correlation analysis was performed on the dataset. The Pearson correlation coefficients between the electricity load and features such as humidity, wind speed, and temperature were calculated. A heat map was generated to depict the relationships between these features visually. As shown in Fig 9, the coefficients for the load with humidity, temperature, and rainfall were 0.69, 0.73, and 0.71, respectively, indicating a strong correlation between the power load and these features.

This study retained the features of temperature, humidity, and rainfall, which demonstrated a strong correlation with load while excluding other features. To assess the impact

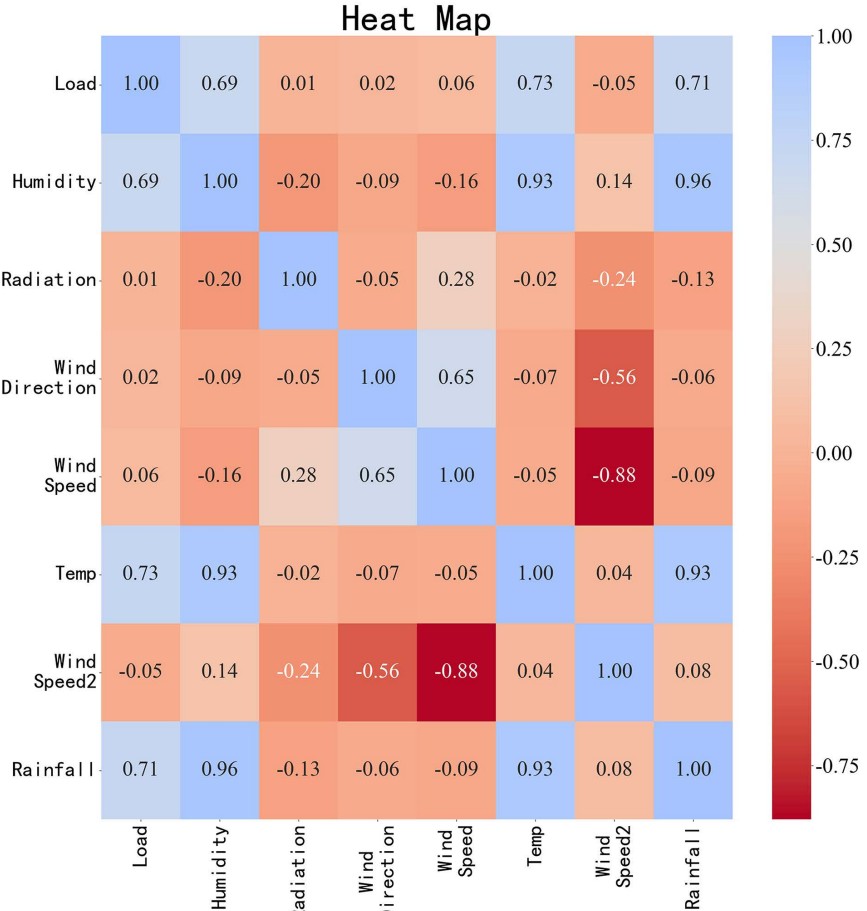

**Fig 9. Correlation Heat Map.**

of the filtered feature set on power load forecasting, a control experiment was conducted to compare the datasets before and after filtering. The XGBoost, LSTM, and TCN models with identical parameter settings were evaluated, and their RMSE, R-squared, and MAE values were recorded.

Fig 10 illustrates the impact of feature engineering on the forecasting results. In the load fitting plot, the vertical axis represents the load values (unit: MWh), while the horizontal axis represents time. This figure depicts the trend of the predicted load values over time and compares them with the actual values. The red line represents the actual load values, the black line shows the predicted results without feature engineering, and the blue line represents the predicted results after applying feature engineering. The three load forecasting plots respectively present the prediction results of the XGBoost, LSTM, and TCN models.

As illustrated in Fig 11, the comparison of the prediction results for the three models across different evaluation metrics is presented. The vertical axis represents RMSE, $R^2$, and MAE, while the horizontal axis corresponds to different models. The figure illustrates the performance of each model in terms of prediction error, goodness of fit, and mean absolute error. As shown in the figure, all three models perform more effectively on the dataset processed using feature engineering. Retaining features with strong correlations to the load data improved the prediction accuracy, emphasizing the importance of temperature, rainfall, and humidity

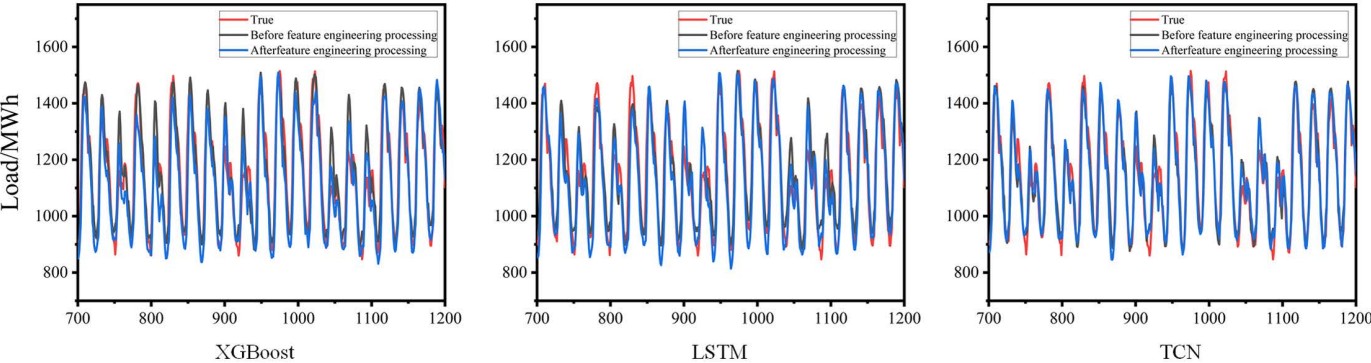

**Fig 10. The Impact of Feature Engineering on Prediction Results.**

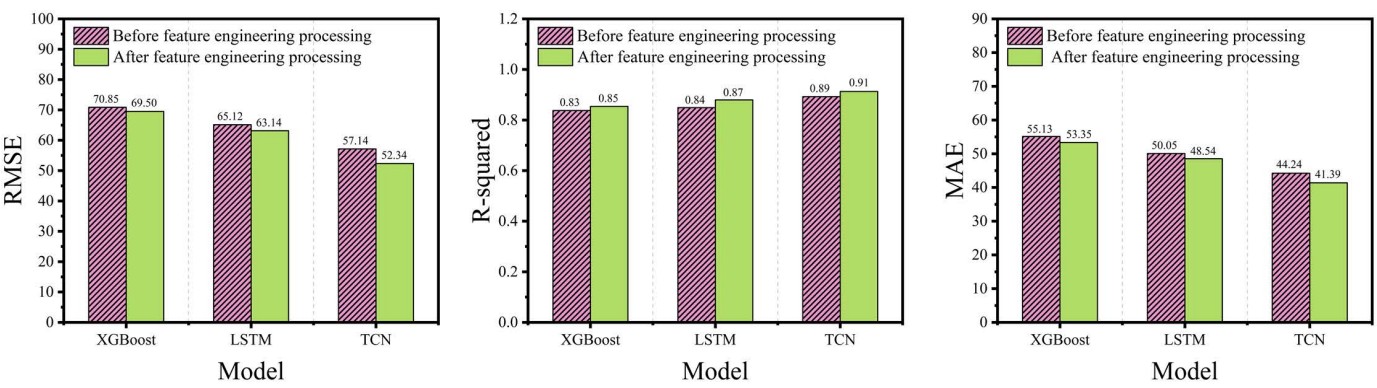

**Fig 11. Comparison of evaluation metrics for the three models.**

for load prediction in this dataset. Among the metrics, TCN presented the greatest improvement in RMSE and MAE, whereas LSTM exhibited the highest improvement in R-squared. These enhancements were attributed to feature engineering, which reduced data and model complexity, as well as resource consumption. The results demonstrated a significant positive impact of feature engineering on the accuracy of the model prediction.

**4.3.3 Deep learning model selection.** To assess the computational efficiency and resource consumption of the BiStacking+TCN-GRU model, the runtime and memory usthe age of various deep learning models were measured under the hardware conditions outlined in Section 4.3.1 Experimental setup. The experimental results are presented in Table 2.

As a CNN variant, the TCN effectively addressed gradient vanishing and explosion issues, enhancing its ability to process complex patterns. Within the RNN family, LSTM captured long-term dependencies using a gating mechanism, whereas BiLSTM improved context understanding and prediction accuracy by incorporating forward and backward information. However, the complex structure of BiLSTM increased computational demands during training and inference. In contrast, GRU had a simplified architecture to accelerate the training process and was appropriate for scenarios with limited computational resources. The experimental results indicated that GRU and BiLSTM required 325.86 and 622.80 s, respectively, to complete 50 epochs, achieving the MAE values of 45.5876 and 71.3103.

The Informer model served as an enhancement of the Transformer model designed for time-series prediction, incorporating a self-attention mechanism that significantly increased

**Table 2. Comparison of resource consumption of deep learning models.**

| Model | Epochs | Time (s) | MAE | Memory Usage (MB) |
|---|---|---|---|---|
| BiLSTM | 50 | 622.80s | 71.3103 | 177.31 |
| BiGRU | 50 | 551.04s | 56.7743 | 163.84 |
| LSTM | 50 | 410.13s | 55.6110 | 81.32 |
| GRU | 50 | 325.86s | 45.5876 | 82.73 |
| Stacking | 50 | **173.44s** | 43.8503 | **79.21** |
| FEDformer | 20 | 1882.18s | 43.0874 | 1228.83 |
| Transformer | 20 | 964.30s | 41.2293 | 1474.56 |
| TCN | 50 | 504.03s | 41.0946 | 245.76 |
| Informer | **20** | 2130.02s | 40.6356 | 2051.22 |
| TCN-GRU | 50 | 810.18s | 37.7232 | 301.48 |
| BiStacking | 50 | 280.77s | 36.5237 | 81.92 |
| Autoformer | 20 | 1501.58s | 35.4563 | 1064.96 |
| BiStacking+GRU | 50 | 596.86s | 31.7659 | 170.39 |
| BiStacking+TCN | 50 | 784.81s | 29.5423 | 346.11 |
| Stacking+TCN-GRU | 50 | 986.51s | 27.4871 | 377.82 |
| **BiStacking+TCN-GRU** | 50 | 1046.74s | **22.5206** | 381.77 |

the computational complexity. The experimental results revealed that, using the same dataset, the Informer model required 2130.02 s to complete 20 epochs, achieving the MAE of 40.6356. In contrast, the proposed model completed the task in 1046.74 seconds, approximately half the runtime of the Informer model. Further comparisons with Transformer, FEDformer, and Autoformer models highlighted that encoder-decoder-based architectures generally entailed longer training times and higher computational demands.

The proposed BiStacking+TCN-GRU model effectively balanced the prediction time, accuracy, and memory usage by reducing the computational overhead and significantly enhancing the prediction accuracy.

**4.3.4 BiStacking+TCN-GRU experimental results.** The proposed model consisted of two components. The first was the double-layer base learner stacking model, referred to as BiStacking, which was an innovative variant of the stacking approach in ensemble learning, classified under machine learning. The second component was a hybrid model combining the TCN and GRU categorized under deep learning.

To evaluate the effectiveness of the proposed model in power load forecasting, independent experiments were first conducted on each component, and their predictive performance was assessed based on the results. Subsequently, the complete model was compared with other models across various categories, and the results were analyzed to verify its prediction accuracy and stability. The experiments used the feature-engineered datasets described in the previous subsection to ensure the objectivity of the evaluation metrics for measuring the performance of the model.

**4.3.4.1 Experiments on BiStacking model prediction:** A comparative experiment was conducted using the classical stacking model to assess its predictive performance. The test set performances of both models are illustrated in Figs 12 and 13.

The load-fitting diagram illustrates the load forecasting results of different models, where the horizontal axis represents time (unit: hours) and the vertical axis represents load (unit: MWh). To provide a clearer comparison of the prediction differences among models over a specific time period, a magnified local view of that period is included in the figure, maintaining the same units for both axes as in the main plot. Additionally, each curve in the figure is

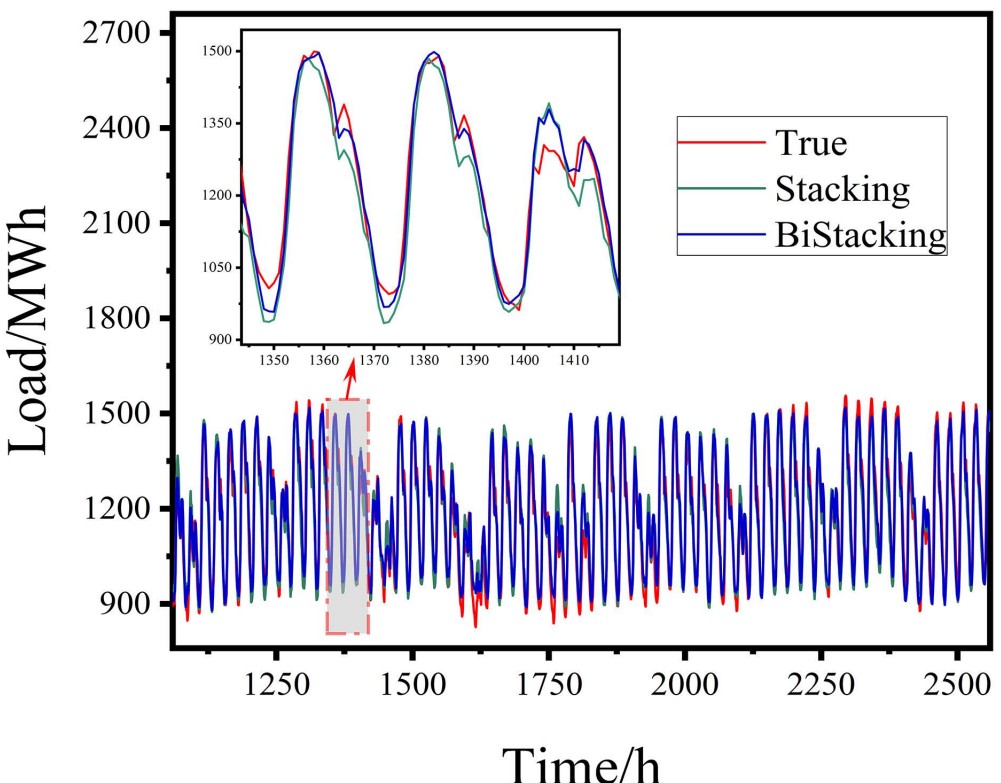

**Fig 12. Prediction Results of the BiStacking Model.**

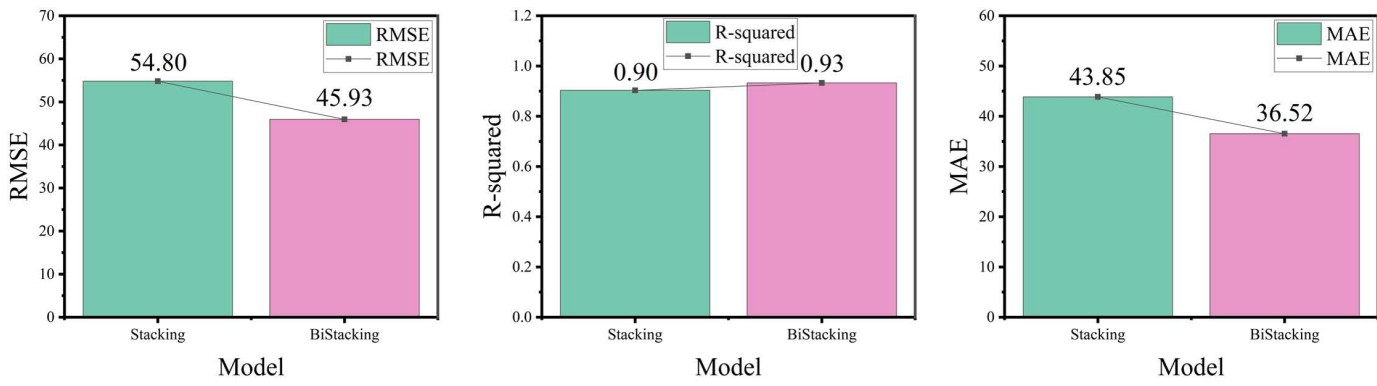

**Fig 13. Evaluation Metrics for the BiStacking Model Experiment.**

labeled with its corresponding model to facilitate the analysis of the performance of different models.

The load-fitting diagram highlighted two key features. First, during brief fluctuations in the actual values amidst a downward trend, the BiStacking model responded more rapidly than the single-layer stacking model, exhibiting a smaller gap between the predicted and actual values. This indicated that BiStacking had a faster response time and maintained higher prediction accuracy during fluctuations. Second, at the load trough, where the actual data began to increase after reaching a minimum, BiStacking provided a more accurate and

timely understanding of the pattern. It adjusted the predictions closer to the subsequent increase, resulting in more precise minimum value predictions. Additionally, as the actual value curve increased at the end of the trough, another form of data fluctuation, the advantages of BiStacking in response time and accuracy further reinforced the observations from Feature 1.

To provide a more intuitive representation of the evaluation metric results and facilitate comparative analysis, this paper presents bar charts illustrating the performance of different models across various metrics. In the charts, the horizontal axis represents the names of the different models, while the vertical axis corresponds to the calculated values of RMSE, $R^2$, and MAE, respectively. Each bar represents the specific calculated value of the corresponding metric, while the overlaid line plot illustrates the trend of these results, reflecting the variations in the metric values, whether increasing or decreasing.

As shown in Fig 13 and Table 3, BiStacking outperformed Stacking in terms of RMSE, R-squared, and MAE. BiStacking achieved an R-squared score 0.0234 higher than that of Stacking, indicating a modest improvement in its ability to explain the data, likely due to the shared application of similar base learners. In terms of RMSE, BiStacking was 8.8753 lower than Stacking, reflecting a significant improvement in error reduction. Additionally, the MAE of BiStacking was 7.3266 lower than that of Stacking, indicating more accurate prediction results.

The performance improvement observed in BiStacking was attributed to its two-layer base-learner configuration. Enhancing the prediction accuracy of stacking methods typically involves optimizing the configuration and the number of base learners. Unlike traditional approaches, this study focused on optimizing the arrangement of base learners to improve the quality of information transfer between them. The two-layer structure in BiStacking achieved superior prediction accuracy compared with the traditional stacking method with similar configurations.

**4.3.4.2 Experiments on TCN-GRU model prediction:** This study analyzed the factors contributing to the improved prediction performance of the TCN-GRU model by comparing it with those of the GRU and TCN models. Detailed comparisons of the prediction results and performance metrics are shown in Figs 14 and 15.

The fitted graph in Fig 14 demonstrated that the TCN-GRU model predicted the minimum and maximum values more accurately than the TCN and GRU models when the true values reached troughs and peaks. Although the TCN and GRU models responded quickly to fluctuations, their predictions were less precise than those of the TCN-GRU model. This indicated that the TCN-GRU model effectively combined the feature extraction capabilities of the TCN with the trend-capturing ability of the GRU, resulting in more accurate predictions during fluctuations.

As shown in Fig 15 and Table 4, the TCN-GRU model surpassed the TCN and GRU models in terms of RMSE, R-squared, and MAE. The results demonstrated that the TCN-GRU model exhibited superior explanatory power and error-reduction capabilities. In addition, its predictions aligned more closely with the true values, achieving an MAE that was 7.8644 lower than that of GRU and 3.3714 lower than that of TCN, thereby highlighting its clear advantage.

**Table 3. Evaluation Metrics for the BiStacking Model Experiment.**

| Model | RMSE | $R^2$ | MAE |
|---|---|---|---|
| Stacking | 54.8080 | 0.9093 | 43.8503 |
| **BiStacking** | **45.9327** | **0.9327** | **36.5237** |

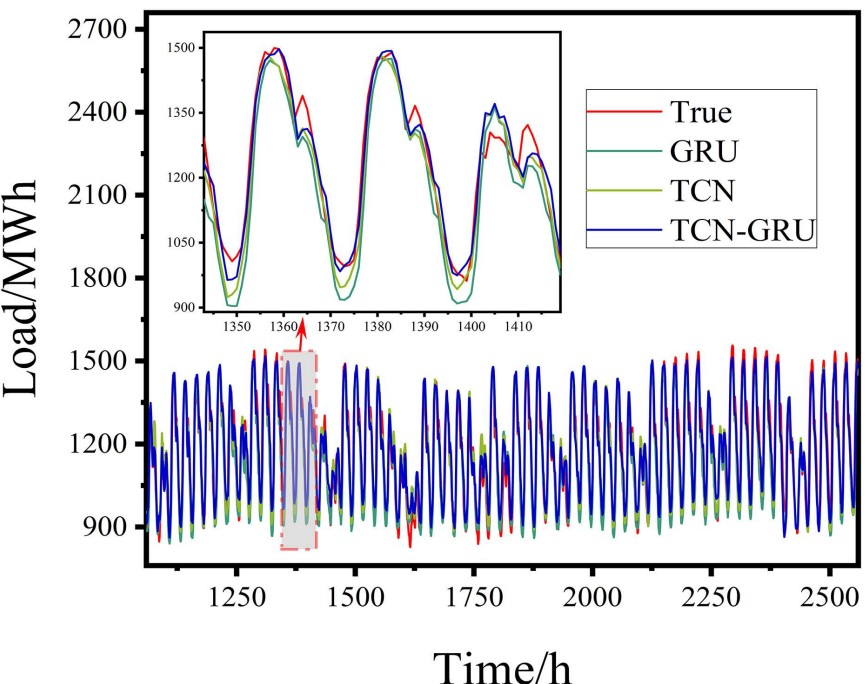

**Fig 14. Prediction Results of the TCN-GRU Model.**

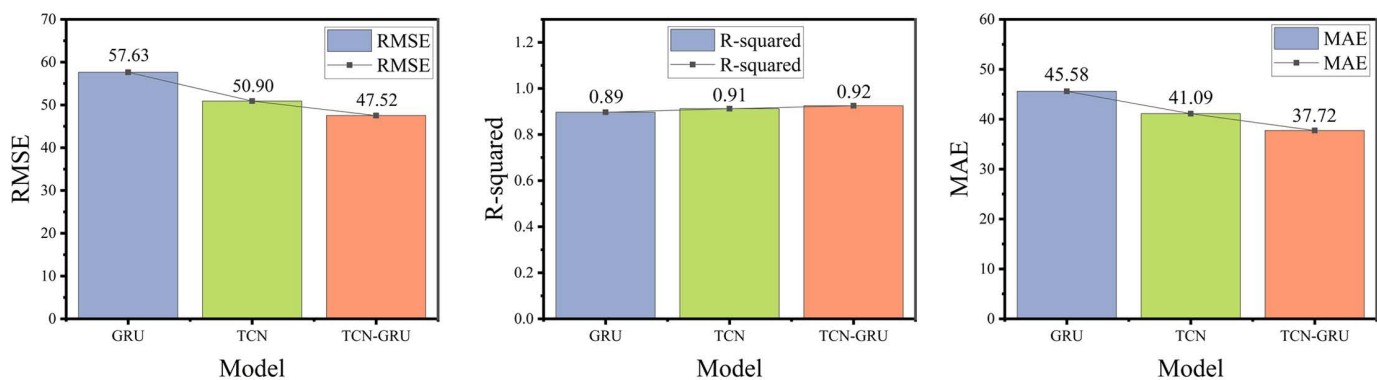

**Fig 15. Evaluation Metrics for the TCN-GRU Model Experiment.**

**Table 4. Evaluation Metrics for the TCN-GRU Model Experiment.**

| Model | RMSE | R² | MAE |
|---|---|---|---|
| GRU | 57.6308 | 0.8968 | 45.5876 |
| TCN | 50.9047 | 0.9123 | 41.0946 |
| **TCN-GRU** | **47.5254** | **0.9287** | **37.7232** |

This improvement was attributed to the complementary principles of TCN and GRU. As an RNN-based network, the GRU employs gating units to capture the long-term dependencies in time series, whereas the TCN serving as a CNN-based network uses an extended convolutional receptive field to analyze the historical data on a macroscopic level. By combining these strengths, the TCN-GRU model enhanced the response speed and stability, resulting in more accurate predictions.

**4.3.4.3 Controlled experiments on all models:** This study presented a comprehensive comparison of all the models examined in the experiments and analyzed their prediction performance on the test set to highlight the differences among various model combinations. Detailed parameter information is provided in Section 4.3.1 Experimental setup, and the test set performance of each model is shown in Figs 16 and 17.

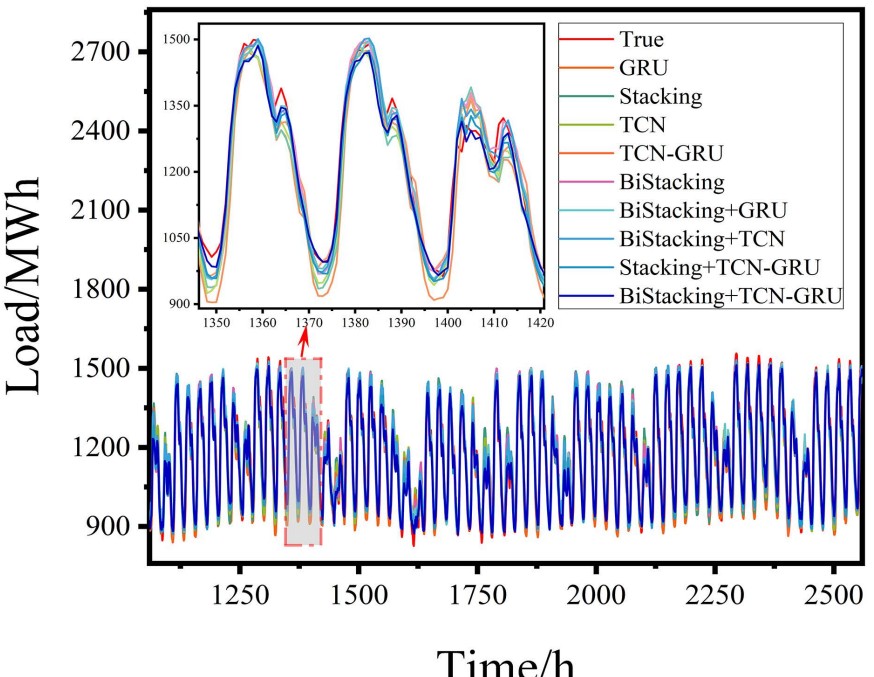

**Fig 16. Predictions of All Models.**

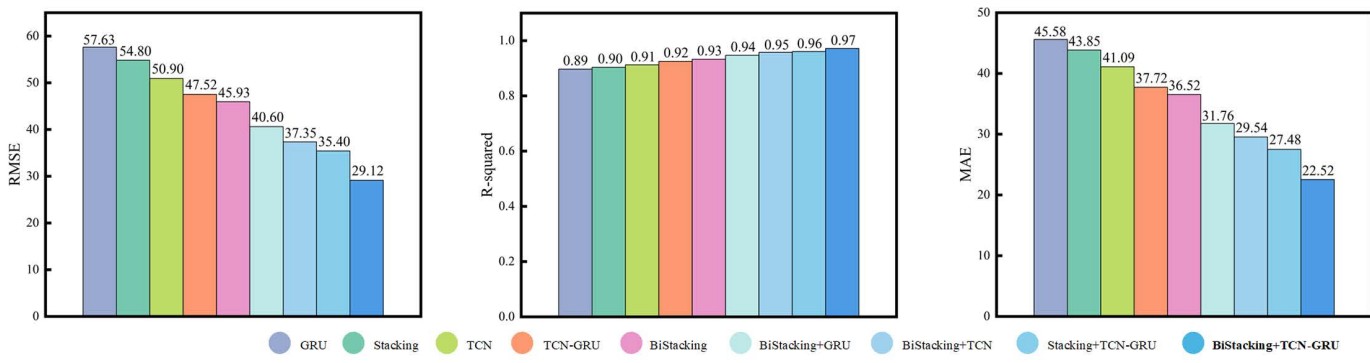

**Fig 17. Evaluation Metrics for All Models.**

Fig 16 illustrates the two key features of the load-fitting results. First, during fluctuations, the BiStacking+TCN and BiStacking+GRU models responded more rapidly and produced predictions closer to the true values than the single models of either the TCN or GRU, demonstrating that the initial processing step significantly enhanced the subsequent prediction accuracy. Second, at troughs or peaks, the BiStacking+TCN-GRU model provided the most accurate predictions among all hybrid models, outperforming both BiStacking+TCN and BiStacking+GRU in alignment with the true values. Furthermore, the Stacking+TCN-GRU model, lacking the two-layer advantage of the BiStacking model, fell short in achieving the accuracy and stability demonstrated by the proposed model.

Fig 17 presents a summary of the evaluation metric results for all models. To facilitate a more intuitive comparison among the models, different colors are used for distinction. As shown in Fig 17 and Table 5, the BiStacking+TCN-GRU model demonstrated significant advantages in terms of RMSE, R-squared, and MAE. With identical parameter configurations, it achieved an R-squared of 0.97, indicating the strongest data interpretation capability. In terms of RMSE and MAE, the model was 16.8114 and 14.0031 higher than BiStacking, which was the best performing among all single models, and 6.2846 and 4.9668 higher than Stacking+TCN-GRU, which was the second-best performing model overall. These results confirmed that the BiStacking+TCN-GRU model provided the predictions closest to the true load values.

The BiStacking model proposed in this study integrated the prediction results of multiple base learners, effectively capturing various patterns in the data while maintaining strong interpretability. By combining the advantages of different models, BiStacking enhanced the ability to recognize and analyze complex patterns, making it more adaptable to data fluctuations and variations. The TCN captured the long-term dependencies in the time-series data, whereas the GRU improved computational efficiency and training stability. By integrating these strengths, the TCN-GRU model overcame the limitations of single deep models, enabling rapid responses to data changes and the enhanced processing of complex time-series data. Consequently, the BiStacking+TCN-GRU model effectively combined the advantages of ensemble and deep learning, excelling in the prediction accuracy, response speed, historical trend analysis, and mitigating the impact of anomalous data.

**4.3.5 Error bar analysis.** To assess the predictive performance of the proposed model, the mean prediction error was plotted to visually compare different models. Fig 18 illustrates the mean prediction error with a 95% confidence interval, where the horizontal axis represents the test models and the vertical axis shows the mean prediction error. A 95% confidence interval indicated that 95% of the prediction errors fell within this range if the

**Table 5. Evaluation metrics for all models.**

| Model | RMSE | R² | MAE |
|---|---|---|---|
| GRU | 57.6308 | 0.8968 | 45.5876 |
| Stacking | 54.8080 | 0.9093 | 43.8503 |
| TCN | 50.9047 | 0.9123 | 41.0946 |
| TCN-GRU | 47.5254 | 0.9287 | 37.7232 |
| BiStacking | 45.9327 | 0.9317 | 36.5237 |
| BiStacking+GRU | 40.6022 | 0.9492 | 31.7659 |
| BiStacking+TCN | 37.3599 | 0.9536 | 29.5423 |
| Stacking+TCN-GRU | 35.4059 | 0.9605 | 27.4871 |
| **BiStacking+TCN-GRU** | **29.1213** | **0.9719** | **22.5206** |

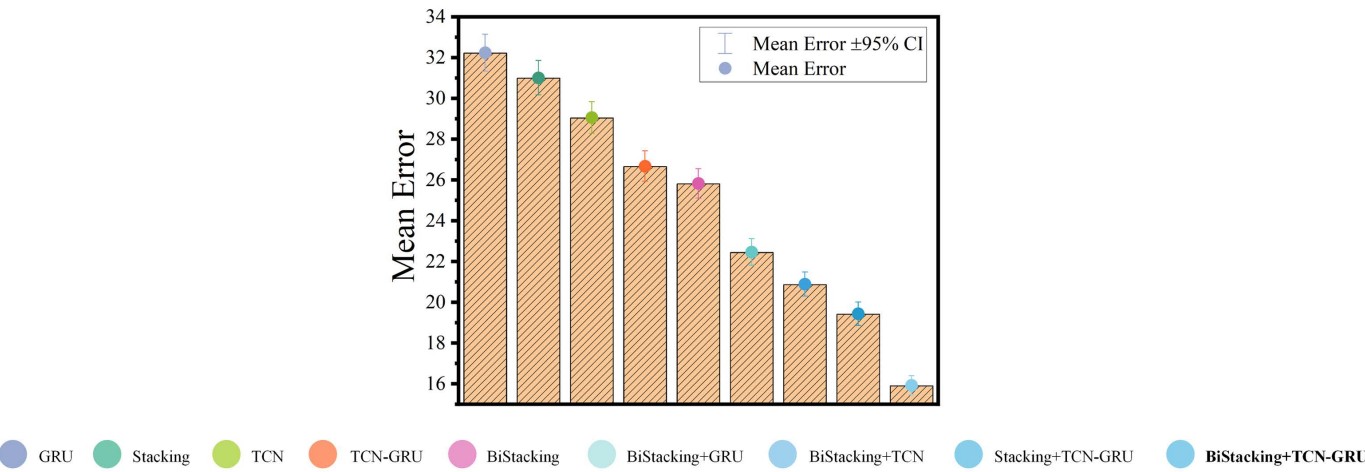

**Fig 18. Error Bar Experiment.**

predictions deviated from the true values. The BiStacking model achieved a low prediction error, significantly enhancing the prediction accuracy.

**4.3.6 Robustness analysis.** Robustness analysis is essential for validating the practical applicability of a model. This study utilized Georgia's hourly electricity load data from January to December 2023, comprising 8,760 data points. The dataset was divided into training and test sets at a 7:3 ratio, with 6,100 samples allocated to the training set and 2,660 to the test set. The prediction results for each model are shown in Figs 19 and 20.

An analysis of the performance metrics, including RMSE, R-squared, and MAE (Fig 20 and Table 6), demonstrated that the proposed BiStacking+TCN-GRU hybrid model consistently delivered strong prediction results across different datasets. This indicated the robustness and suitability of the model for power prediction tasks in diverse application scenarios.

## 5 Conclusion and future research directions

This study proposed a hybrid forecasting model of BiStacking+TCN-GRU, which integrated machine learning and deep learning algorithms for short-term power load forecasting. The PCC algorithm was first applied to select features with a high correlation to power loads. Using these features, the BiStacking model generated a preliminary prediction, which served as an input for the TCN-GRU model. The TCN-GRU model enhanced the ability to capture time-series features, thereby improving predictive performance. The final output was derived from the predictions of the TCN-GRU model.

The model employed distributed and small-batch training strategies to efficiently manage large-scale datasets and high-dimensional feature spaces, thereby reducing memory consumption and accelerating convergence. Dimensionality reduction was applied to the input features to further alleviate computational and memory overheads. K-fold cross-validation was used to partition the dataset and to mitigate overfitting. In addition, data normalization enhanced stability and improved the prediction accuracy during training.

The following conclusions were drawn from the experiments and analyses:

(1) Feature engineering could be an essential step in power load forecasting, as effective feature selection significantly enhanced model prediction accuracy. The application of the PCC algorithm eliminated features with low correlation to the power load, reducing the computational redundancy and lowering the computational costs.

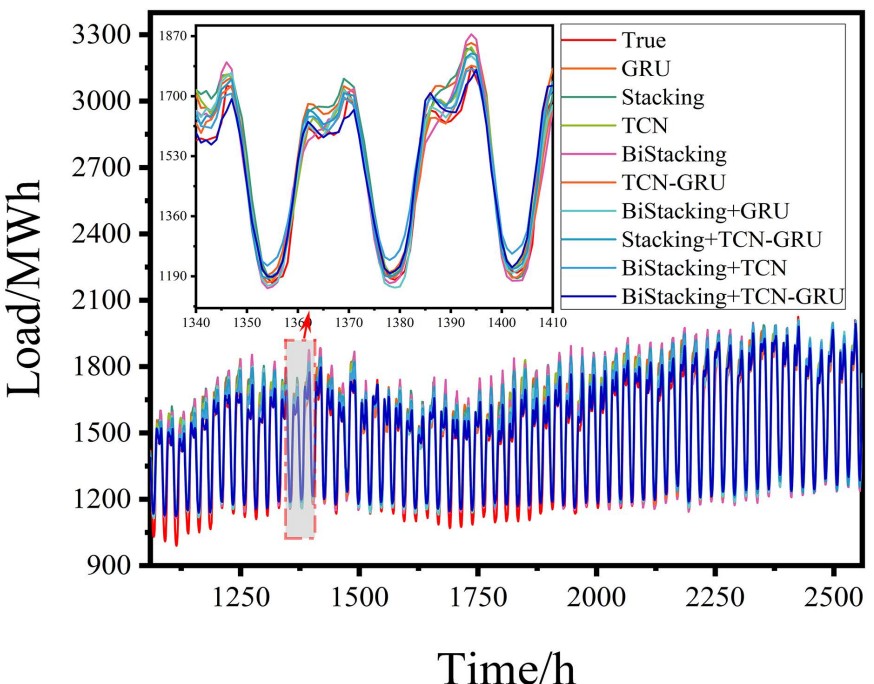

**Fig 19. Results of the robustness experiment.**

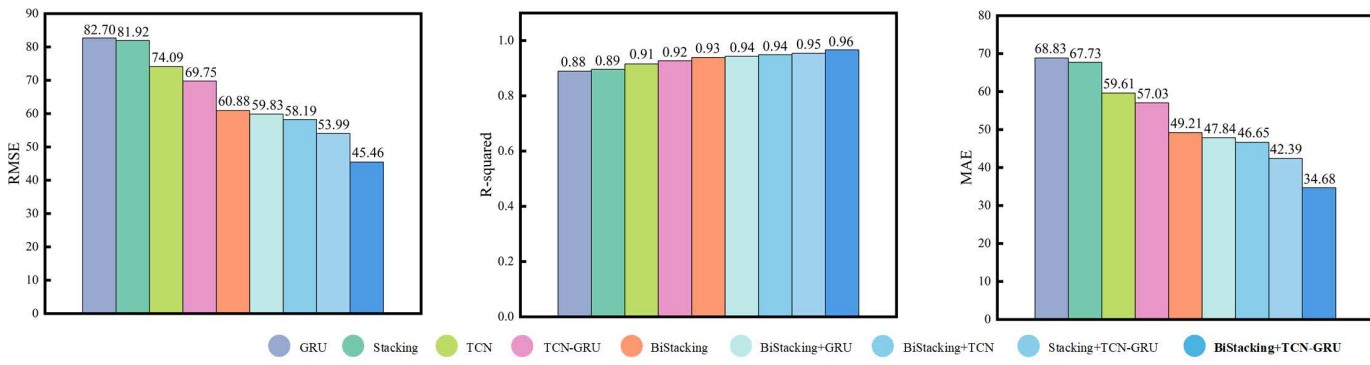

**Fig 20. Evaluation metrics of the robustness experiment.**

**Table 6. Evaluation metrics for the robustness experiment.**

| Model | RMSE | R² | MAE |
|---|---|---|---|
| GRU | 82.7035 | 0.8891 | 68.8349 |
| Stacking | 81.9219 | 0.8960 | 67.7348 |
| TCN | 74.0917 | 0.9156 | 59.6190 |
| BiStacking | 69.7557 | 0.9271 | 57.0336 |
| TCN-GRU | 60.8804 | 0.9391 | 49.2171 |
| BiStacking+GRU | 59.8394 | 0.9433 | 47.8435 |
| Stacking+TCN-GRU | 58.1979 | 0.9487 | 46.6574 |
| BiStacking+TCN | 53.9991 | 0.9539 | 42.3983 |
| **BiStacking+TCN-GRU** | **45.4616** | **0.9665** | **34.6897** |

(2) The proposed BiStacking model enhanced the ability to recognize and analyze complex data patterns through its two-layer base learner structure. Compared with the traditional stacking model, BiStacking extracted more comprehensive information from the data, offering greater flexibility in selecting the number and types of base learners. This flexibility enabled the model to achieve better performance in terms of prediction accuracy and generalization ability.

(3) The TCN-GRU model integrated the feature extraction capabilities of the TCN with the GRU's ability to capture sequence trends, resulting in strong predictive performance in power load forecasting tasks. Although previous research on hybrid models combining ensemble learning and deep learning has demonstrated certain limitations, the BiStacking+TCN-GRU model proposed in this study offered a more comprehensive approach to capturing time-series information than other deep learning hybrid algorithms. This model introduced a novel method for combining ensemble learning with deep learning.

(4) The proposed model achieved RMSE, R-squared, and MAE values of 29.1213, 0.9719, and 22.5206, respectively, outperforming other models. Compared to the suboptimal Stacking+TCN-GRU model, the RMSE was reduced by 17.7%, the R-squared was improved by 1.04%, and the MAE was reduced by 18%. Additionally, robustness experiments demonstrated the superior accuracy and generalization ability of the proposed hybrid model, with RMSE, R-squared, and MAE values of 45.4616, 0.9665, and 34.6897, respectively.

The BiStacking+TCN-GRU model demonstrated excellent performance in power load forecasting; however, it still had certain limitations. While the model was adaptable to different regions, its predictive performance could be affected under extreme weather conditions or unexpected events. Therefore, to address the challenges of complex and dynamic operational environments, this study integrated offline training with online prediction methods and proposed a model update strategy with a correction mechanism and model replication strategy specifically designed for extreme weather events.

During model deployment, a prediction accuracy threshold was set based on practical application requirements. When the accuracy of the model stabilized within the threshold range, it was manually updated at regular intervals by retraining it with the latest data and historical data from a specified time range, ensuring the model effectively captures recent trends and improves its adaptability to dynamic environments. The correction mechanism is triggered if the prediction accuracy falls below the threshold more than 10 times within a fixed interval, prompting immediate retraining until the accuracy returns within the threshold. The fixed interval was then recalculated from the correction point to maintain the continuity and stability of the update strategy.

The model replication strategy addressed the challenges of load forecasting under extreme weather conditions. During the first occurrence of an extreme weather event, load and meteorological data were manually recorded and used to train a model replica tailored to the specific weather type. For subsequent similar events, the corresponding predictive model was manually selected to improve forecasting accuracy under extreme weather conditions.

Regarding the generalization ability of the model, experimental results demonstrated that the proposed model performs excellently in power load forecasting and exhibits strong predictive capabilities. For similar load forecasting tasks in different regions, the model can effectively adapt and generate predictions as long as historical load data from the target region is provided. Additionally, in terms of handling different time periods, the model adopts a manual update strategy that can be flexibly adjusted to accommodate various forecasting time requirements, thereby enhancing its generalizability and application scope. Although this study did not directly involve real-world deployment, the experimental results validated the

model's effectiveness in power load forecasting, highlighting its potential for practical applications. Future research will further explore deployment strategies in real-world scenarios and optimize the model to meet diverse requirements, ultimately facilitating its integration into power grid management and promoting its practical implementation.

The proposed hybrid prediction model is essential for short-term load forecasting and offers a practical and reliable approach to load forecasting for optimal power dispatching in the power industry. Its simple deployment and ease of maintenance render it particularly advantageous for large-scale applications and long-term operations, thereby supporting the development of advanced power dispatching strategies.

Future research will focus on several key areas to enhance the capabilities of the BiStacking+TCN-GRU model:

(1) Enhancing the Ability to Process Large-Scale Datasets: As the volume of data increases, processing large-scale datasets becomes a critical challenge. To address this, the BiStacking+TCN-GRU model could explore integrating incremental learning, structural pruning techniques, and optimization algorithms to adjust parameters. These approaches would improve training efficiency and enable more effective handling of large-scale datasets.

(2) Effective Fusion of Multi-Feature Data: Another important direction is the effective fusion of multi-feature data, particularly considering the influence of factors such as weather, energy storage, and grid capacity. By incorporating attention mechanisms, the model could assign appropriate weights to these factors, thereby improving its adaptability and robustness.

(3) Expanding the Forecasting Capability for Long-Term Trends: Given the increasing demand for mid- and long-term power load forecasting, expanding the prediction horizon and enhancing the model's ability to capture long-term trends and cyclical fluctuations is essential. The BiStacking+TCN-GRU model could benefit from combining multi-scale temporal convolutions with periodic feature extraction methods, which would enhance its capacity to model long-term patterns and better meet the needs of long-term load forecasting.

(4) Enhancing the Handling of Anomalous Data: The presence of noise or incomplete data can negatively impact prediction results. To improve the model's ability to handle such anomalies, outlier detection and noise filtering algorithms can be employed to clean noise and anomalies in the data. Additionally, integrating signal decomposition techniques can help effectively separate noise components from trend components within the signal, thereby enhancing the model's prediction accuracy and robustness.

## Author contributions

**Conceptualization:** Jun Ma, Jishen Peng, Haotong Han, Liye Song.

**Data curation:** Jun Ma, Haotong Han.

**Formal analysis:** Jun Ma, Haotong Han.

**Investigation:** Hao Liu.

**Methodology:** Jun Ma.

**Software:** Jun Ma.

**Visualization:** Jun Ma.

**Writing – original draft:** Jun Ma, Haotong Han.

**Writing – review & editing:** Jun Ma, Hao Liu.

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
