## [Decision Letter · Decision Letter 0]

20 Oct 2024

PONE-D-24-40595A hybrid power load forecasting model combining ensemble learning and deep learning: BiStacking+TCN-GRUPLOS ONE

Dear Dr. Ma,

<please by="" manuscript="" revised="" submit="" your="">Thank you for submitting your manuscript to PLOS ONE. After careful consideration, we feel that it has merit but does not fully meet PLOS ONE’s publication criteria as it currently stands. Therefore, we invite you to submit a revised version of the manuscript that addresses the points raised during the review process.</please> Please submit your revised manuscript by Dec 04 2024 11:59PM. If you will need more time than this to complete your revisions, please reply to this message or contact the journal office at plosone@plos.org. <please by="" manuscript="" revised="" submit="" your="">

Please include the following items when submitting your revised manuscript:</please>

We look forward to receiving your revised manuscript.

Kind regards,

Anurag Sinha, Ph.D

Academic Editor

PLOS ONE

Journal Requirements:

2. Please note that PLOS ONE has spec6ific guidelines on code sharing for submissions in which author-generated code underpins the findings in the manuscript. In these cases, all author-generated code must be made available without restrictions upon publication of the work. Please review our guidelines at https://journals.plos.org/plosone/s/materials-and-software-sharing#loc-sharing-code and ensure that your code is shared in a way that follows best practice and facilitates reproducibility and reuse.

Additional Editor Comments:

There are some minor grammatical issues and awkward phrasing that could be improved with additional proofreading to enhance readability.

-Data are plural

- In abstract: line 51 the system integrates automation with real-time data processing >> the system employes automation with real-time data processing

Reviewers' comments:

Reviewer's Responses to Questions

**Comments to the Author**

1. Is the manuscript technically sound, and do the data support the conclusions?

Reviewer #1: Yes

Reviewer #2: Yes

2. Has the statistical analysis been performed appropriately and rigorously? 

Reviewer #1: Yes

Reviewer #2: Yes

3. Have the authors made all data underlying the findings in their manuscript fully available?

Reviewer #1: Yes

Reviewer #2: Yes

4. Is the manuscript presented in an intelligible fashion and written in standard English?

Reviewer #1: Yes

Reviewer #2: Yes

5. Review Comments to the Author

Reviewer #1: Questions related to the conducted research in this paper:

- How might the model's reliance on historical data influence its performance during unprecedented load variations,

such as during extreme weather events?

- What specific features did the PCC algorithm identify as crucial for improving prediction accuracy in power load forecasting?

- How does the performance of the BiStacking+TCN-GRU model compare to other state-of-the-art models in the literature on short-term load forecasting?

Comments/improvements related to overall structure of the paper and diction used:

- Ensure a consistent writing style, as some sections are more technical, while others are relatively casual or less formal.

- In the introduction and conclusion, emphasize the practical applications and real-world impact of the proposed model to align the technical work with practical significance.

- Reorganize some of the experiment subsections to prevent overlapping descriptions, ensuring each section introduces new insights or findings.

- Include a comparison of model training times and computational costs for each model to evaluate trade-offs between performance and resource consumption.

- Justify the selection of TCN and GRU over other deep learning models or hybrid combinations, explaining why this combination was chosen.

Reviewer #2: In this paper, the authors propose a novel hybrid model combining ensemble learning and deep learning techniques for power load forecasting. The BiStacking+TCN-GRU model leverages the strengths of both approaches to enhance prediction accuracy in short-term load forecasting (STLF). While the study addresses the important challenge of improving load forecasting in the power industry, several suggestions can be made to enhance the clarity and depth of the methodology before recommending it for publication:

1. How does the integration of BiStacking with TCN-GRU specifically improve predictive accuracy compared to traditional forecasting methods?

2. I would highly suggest the authors to consolidate repetitive information, especially when discussing the results of multiple models, to avoid redundancy and enhance clarity.

3. The authors should try to elaborate more on how the model addresses potential issues related to overfitting, especially given the complexity introduced by the BiStacking approach.

4. Although the graphs effectively illustrate the performance comparisons. I would recommend the authors to add error bars or statistical significance tests would strengthen the results.

5. How does this approach compare to other dynamic updating methods for rough sets or related models (e.g., fuzzy rough sets, probabilistic rough sets)?

6. Could this model be adapted for streaming data scenarios where objects are continuously added or removed?

7. The Preliminaries section is generally helpful for establishing key definitions and concepts. Therefore, I would suggest the authors to consider moving some of the more technical details to methodlogical sections to improve readability for other researchers.

8. The authors should mention the challenges and potential approaches for scaling this model to very large datasets or high-dimensional feature spaces.

6. PLOS authors have the option to publish the peer review history of their article (what does this mean? ). If published, this will include your full peer review and any attached files.

**Do you want your identity to be public for this peer review?** For information about this choice, including consent withdrawal, please see our Privacy Policy .

Reviewer #1: No

Reviewer #2: **Yes: ** Shraiyash Pandey

---

## [Author Response · Author response to Decision Letter 0]

3 Dec 2024

Response to comments by Editors

Thank you for your valuable feedback. We have thoroughly reviewed the document, making the necessary revisions to enhance readability and address all minor grammatical issues. Additionally, we have ensured that data is consistently used in its correct plural form throughout the document.

Response to comments by Reviewer #1

Comments

1. How might the model's reliance on historical data influence its performance during unprecedented load variations, such as during extreme weather events

Response:

Regarding the model's performance in extreme weather events, we integrated meteorological data (such as temperature and humidity) with historical load data to enhance the BiStacking+TCN-GRU model's adaptability to load fluctuations.

The model primarily handles data under normal weather conditions during routine operation. However, to address extreme weather events that may occur (such as heavy rainfall or snowfall), a model replication strategy is employed to manage these special cases. Specifically, when an extreme weather event is encountered for the first time, load and meteorological data under these conditions are manually recorded, and these data are then used to train a model replica for that specific type of weather. This approach minimizes the influence of normal data on the prediction results for extreme weather by constructing a dedicated model tailored to such extreme conditions.

When similar weather events recur, the corresponding predictive model is manually selected to perform load forecasting, thereby enhancing forecasting accuracy under extreme weather conditions. As data on extreme weather events continues to accumulate, the model replicas are iteratively updated, further improving their generalization capabilities and predictive accuracy across diverse weather scenarios.

To provide a clearer understanding of the strategies employed for addressing extreme weather events, we have elaborated on the relevant details in the conclusion.

2. What specific features did the PCC algorithm identify as crucial for improving prediction accuracy in power load forecasting?

Response:

We have provided a detailed description of the key features identified by the PCC algorithm in Section 4.3.2 of the revised manuscript. Our analysis indicates that temperature, humidity, and rainfall are crucial factors influencing the prediction accuracy of the electricity load forecasting model. This additional information has been incorporated into the manuscript to enhance the clarity of the factors that impact the model's performance.

3. How does the performance of the BiStacking+TCN-GRU model compare to other state-of-the-art models in the literature on short-term load forecasting?

Response:

We conducted additional experiments to compare our model with four leading algorithms: Transformer, Informer (DOI: 10.1609/aaai.v35i12.17325), FEDformer (DOI: 10.48550/arXiv.2201.12740), and Autoformer (DOI: arxiv-2106.13008). These models were evaluated in terms of prediction accuracy, training time, and memory usage, with the results presented in Table 2 in Section 4.3.3 of the revised manuscript. Furthermore, a comparison of our model with the most advanced models currently available is also provided in Table 1.

Table 1. Comparison of resource consumption of deep learning models

Model Epochs Time(s) MAE Memory Usage (MB)

GRU 50 325.86s 45.5876 82.73

FEDformer 20 1882.18s 43.0874 1228.83

Transformer 20 964.30s 41.2293 1474.56

TCN 50 504.03s 41.0946 245.76

Informer 20 2130.02s 40.6356 2051.22

TCN-GRU 50 810.18s 37.7232 301.48

Autoformer 20 1501.58s 35.4563 1064.96

BiStacking+TCN-GRU 50 1046.74s 22.5206 381.77

According to the analysis of the table data, models based on the encoder-decoder architecture require longer training times and higher computing resource requirements due to their structural complexity. In addition, models based on the encoder-decoder architecture and attention mechanisms typically have high hardware requirements during deployment and necessitate specialized training for staff in operation and maintenance, which significantly increases implementation costs. In contrast, the BiStacking+TCN-GRU model not only substantially reduces computational resource consumption and training time but also maintains high prediction accuracy. It is straightforward to operate and maintain, ensuring seamless workflow transitions and stable system performance even in the event of changes in operation or maintenance personnel. These advantages make it well-suited for rapid deployment and long-term operation in practical engineering applications.

4. Ensure a consistent writing style, as some sections are more technical, while others are relatively casual or less formal.

Response:

We greatly appreciate the valuable feedback, which has significantly contributed to improving the quality of our paper. In response to this feedback, we have made the necessary revisions to ensure a more consistent, formal, and technical writing style throughout the manuscript and we have standardized the language to adopt a more formal tone in sections that were previously less formal.

5. In the introduction and conclusion, emphasize the practical applications and real-world impact of the proposed model to align the technical work with practical significance.

Response:

We acknowledge the importance of integrating technical details with practical applications. In response to your feedback, we have revised the "Introduction" and "Conclusion" sections of the manuscript to more explicitly highlight the practical applications of the BiStacking+TCN-GRU model.

Introduction: We have included a discussion on the role of electricity load forecasting in optimizing grid operations and power dispatching at the beginning of the Introduction. This addition underscores the critical importance of forecasting for maintaining grid stability and ensuring the efficient distribution of energy.

End of the Introduction: In the revised conclusion of the Introduction, we highlight the practical applications of the BiStacking+TCN-GRU model and explore its potential to enhance power load forecasting and improve grid stability.

Conclusion: In the Conclusion, we emphasize the practical applications of the proposed model, particularly its advantages in reducing the demand for computational resources and training time, which makes it more suitable for industrial deployment. Additionally, we highlight the model’s operational stability and maintainability, both of which are crucial for its large-scale implementation in real-world scenarios.

6. Reorganize some of the experiment subsections to prevent overlapping descriptions, ensuring each section introduces new insights or findings.

Response:

We fully agree with your suggestion and have removed the separate experimental section for the Stacking+TCN-GRU model, integrating its findings and conclusions into the comparative analysis of all models. Through this reorganization, we have presented the experimental results more coherently, ensuring that each section introduces new insights and findings, thereby avoiding redundancy. These revisions have enhanced the clarity and fluency of the manuscript, while more effectively highlighting the comparative performance of the proposed model relative to others.

7. Include a comparison of model training times and computational costs for each model to evaluate trade-offs between performance and resource consumption.

Response:

We acknowledge the importance of evaluating the trade-offs between model performance and resource consumption to assess the practical feasibility of different models. To address this, we conducted additional experiments comparing the training time and computational costs of each model. The relevant results are presented in Table 2, Section 4.3.3 of the revised manuscript. We believe that this addition enhances the manuscript by providing a more comprehensive evaluation of both the predictive performance and resource efficiency of the models.

8. Justify the selection of TCN and GRU over other deep learning models or hybrid combinations, explaining why this combination was chosen.

Response:

We have conducted a comparative analysis of various deep learning models, which is presented in Section 4.3.3 of the revised manuscript. The decision to combine TCN and GRU was made based on the following considerations:

TCN’s Ability to Handle Complex Patterns: As a variant of CNN, TCN effectively addresses the vanishing and exploding gradient problems commonly encountered in traditional RNNs. This enables TCN to capture long-range dependencies in sequential data while maintaining both stability and efficiency in processing complex patterns.

GRU’s Efficiency in Sequential Learning: In comparison to LSTM and BiLSTM, GRU simplifies the network architecture by reducing the number of gates and parameters, thereby improving training speed and making it more suitable for environments with limited computational resources.

Comparison to Transformer Models: Transformer-based models require significant computational resources, particularly during the training process. Additionally, the complexity of these models makes them computationally expensive and difficult to deploy in environments with limited resources.

By combining TCN with GRU, our model leverages the strengths of both CNN and RNN architectures while avoiding the computational overhead associated with more complex models, such as BiLSTM and Transformer-based networks. This hybrid approach achieves a balance between predictive accuracy, training efficiency, and computational resource consumption, making it particularly suitable for practical applications where computational resources and time are limited.

Response to comments by Reviewer #2

Comments

1. How does the integration of BiStacking with TCN-GRU specifically improve predictive accuracy compared to traditional forecasting methods?

Response:

To clarify that the improvement in predictive accuracy is primarily driven by the integration of BiStacking with TCN-GRU, we have provided additional details in Section 4.3.4, Experiment 3, of the revised manuscript.

The Tandem Base Learner in the BiStacking model enhances feature learning capabilities and effectively integrates the advantages of multiple Tandem Base Learners in the bidirectional selection layer. This improves the generalization ability of the model and enhances its ability to recognize complex data patterns.

TCN excel in capturing long-range dependencies and intricate temporal patterns. Compared to RNN, TCN effectively mitigate vanishing and exploding gradient issues, thereby substantially improving the model's capability to handle extended sequences.

The GRU employs a simplified gating mechanism that efficiently captures both short-term and long-term dependencies, while offering greater computational efficiency and faster training compared to other RNN architectures, such as LSTM.

By combining BiStacking with the TCN-GRU model, our approach efficiently integrates the prediction results of multiple Tandem Base Learners and fully leverages the advantages of ensemble learning. The robust sequential modeling capabilities of both TCN and GRU enable the BiStacking+TCN-GRU model to accurately capture the time-dependent and dynamic characteristics of the data, thereby significantly enhancing its ability to process complex time-series data, respond quickly to data fluctuations, and improve prediction accuracy and reliability in practical applications.

2. I would highly suggest the authors to consolidate repetitive information, especially when discussing the results of multiple models, to avoid redundancy and enhance clarity.

Response:

We fully agree with your recommendation, as it significantly enhances the clarity and coherence of the manuscript. In response, we have consolidated the results of Experiments 3 and 4 in Section 4.3.4 into a single, unified experiment. This revision reduces redundancy and streamlines the presentation of the results.

Additionally, we have removed redundant sections of the manuscript to enhance overall clarity and expression.

3. The authors should try to elaborate more on how the model addresses potential issues related to overfitting, especially given the complexity introduced by the BiStacking approach.

Response:

Your suggestions are invaluable in improving the manuscript. In response, we elaborate in Section 3.3 on the K-fold cross-validation and data normalization methods employed in the model to mitigate overfitting.

K-fold Cross-validation: By employing K-fold cross-validation, we ensure that the model is evaluated on multiple subsets of the data, thereby mitigating overfitting. This approach ensures that the model's performance is not overly reliant on any single training set and provides a more robust estimate of its generalization ability.

Data Normalization: We apply data normalization to standardize the input features, ensuring that they are on a comparable scale. This enhances the model's training stability and further reduces the risk of overfitting.

In summary, the BiStacking approach effectively addresses the potential for overfitting and enhances the robustness and generalization of the model through the integration of K-fold cross-validation and data normalization techniques.

4. Although the graphs effectively illustrate the performance comparisons. I would recommend the authors to add error bars or statistical significance tests would strengthen the results.

Response:

We fully agree with the suggestion to include error bars, and have added them to the graphs in Section 4.3.5 to more clearly illustrate the performance of the models. The inclusion of error bars enhances the presentation by demonstrating the stability and reliability of the results, particularly when comparing the performance of different models.

5. How does this approach compare to other dynamic updating methods for rough sets or related models (e.g., fuzzy rough sets, probabilistic rough sets)?

Response:

In our study, the BiStacking+TCN-GRU model combines the strengths of ensemble learning and deep learning. Unlike methods such as fuzzy rough sets and probabilistic rough sets, which primarily address uncertainty and fuzzy data, our model focuses on capturing the complex temporal patterns and both long- and short-term dependencies within electricity load data. As a TCN, it automatically learns multi-level features from time-series data, while the GRU effectively captures long-term dependencies through its gating mechanism. Therefore, our model excels in time-series modeling.

While methods like fuzzy rough sets and probabilistic rough sets are effective in handling noisy and incomplete data, they often require additional strategies to maintain high predictive performance when dealing with large volumes of time-series data. In contrast, our BiStacking+TCN-GRU model, leveraging the robustness of ensemble learning and the nonlinear modeling capabilities of deep learning, effectively handles noisy and incomplete data in electricity load forecasting, while maintaining high prediction accuracy.

To adapt to the complex and dynamic operating environment, this study designs a model normal update method that includes a correction mechanism, enabling rapid adaptation to changing operating conditions and load fluctuations when environmental conditions change significantly. Under extreme weather conditions, prediction model replicas specifically trained for such conditions is invoked to effectively address power load fluctuations caused by weather variations. Unlike approaches such as fuzzy rough sets and probabilistic rough sets, which rely on static rules and fixed data processing methods, our model achieves high prediction accuracy in dynamically changing load data through a flexible update strategy.

6. Could this model be adapted for streaming data scenarios where objects are continuously added or removed?

Response:

In order to adapt to the complex and ever-changing operating environment, this study combines offline training with online prediction methods to design a model update approach that includes a correction mechanism. This approach not only enables the model to ad

---

## [Decision Letter · Decision Letter 1]

25 Dec 2024

PONE-D-24-40595R1A hybrid power load forecasting model combining ensemble learning and deep learning: BiStacking+TCN-GRUPLOS ONE

Dear Dr. Ma,

Thank you for submitting your manuscript to PLOS ONE. After careful consideration, we feel that it has merit but does not fully meet PLOS ONE’s publication criteria as it currently stands. Therefore, we invite you to submit a revised version of the manuscript that addresses the points raised during the review process.

We look forward to receiving your revised manuscript.

Kind regards,

Anurag Sinha, Ph.D

Academic Editor

PLOS ONE

Journal Requirements:

Reviewers' comments:

Reviewer's Responses to Questions

**Comments to the Author**

1. If the authors have adequately addressed your comments raised in a previous round of review and you feel that this manuscript is now acceptable for publication, you may indicate that here to bypass the “Comments to the Author” section, enter your conflict of interest statement in the “Confidential to Editor” section, and submit your "Accept" recommendation.

Reviewer #1: (No Response)

Reviewer #2: (No Response)

2. Is the manuscript technically sound, and do the data support the conclusions?

Reviewer #1: Yes

Reviewer #2: Yes

3. Has the statistical analysis been performed appropriately and rigorously? 

Reviewer #1: Yes

Reviewer #2: Yes

4. Have the authors made all data underlying the findings in their manuscript fully available?

Reviewer #1: Yes

Reviewer #2: Yes

5. Is the manuscript presented in an intelligible fashion and written in standard English?

Reviewer #1: Yes

Reviewer #2: Yes

6. Review Comments to the Author

Reviewer #1: The research conducted shows good results, however, the formatting of the research paper as well as the presentation itself is not up to the mark. A few considerations are:

1. There are two subsection headings that include no overview about them, subsection 2.2 and subsection 4.3. Both the subsections don't provide any details about the section itself and move forward with it's subsequent subsections. I'd recommend the authors to fix this to improve readability and comprehension.

2. Subsection 2.1 on "Data analysis and feature engineering" provides almost very little information about the topic. Although, it's a subsection, it should provide a more in-depth discussion on the Data analysis and feature engineering.

3. Inclusion of a literature review is needed. The research has conducted a comparison study, however, without a proper literature review among other works, it's very hard to judge the emphasize of the results.

4. A proper section dedicated to further future works will enhance the discussion presented within the paper.

Reviewer #2: The authors have done a great job in re-constructing the paper based on the last revision comments. However, there are still a few things I'd like to add a few comments.

1. Title the Conclusion section "Conclusion and Future Research Directions" as it includes a few details about the future work as well.

2. Subsection 4.3 has no information about it's heading, it directly jumps into 4.3.1 and so on. This shows inconsistent flow of information. Add neccessary information about each heading instead of leaving it blank.

3. Provide more information about the evaluation metrics. The information provided is not adaequte enough.

4. Perhaps a related work section could improve the quality of the research paper and add more depth and weight to the research itself

7. PLOS authors have the option to publish the peer review history of their article (what does this mean? ). If published, this will include your full peer review and any attached files.

**Do you want your identity to be public for this peer review?** For information about this choice, including consent withdrawal, please see our Privacy Policy .

Reviewer #1: **Yes: ** Surendra Pandey

Reviewer #2: No

---

## [Author Response · Author response to Decision Letter 1]

3 Jan 2025

Response to comments by Editors:

Thank you for your valuable feedback. We have thoroughly reviewed the reference list and confirmed that all citations are complete and correct. Upon inspection, we found no retracted papers in the reference list.

Response to comments by Reviewer #1:

Comments:

1. There are two subsection headings that include no overview about them, subsection 2.2 and subsection 4.3. Both the subsections don't provide any details about the section itself and move forward with it's subsequent subsections. I'd recommend the authors to fix this to improve readability and comprehension.

Response:

Thank you for your valuable feedback. In response to your comment regarding the lack of an overview in subsections 2.2 and 4.3, we have added introductory paragraphs at the beginning of both sections to provide a clearer summary of their content. In subsection 2.2, we describe the base models and algorithms used in this study, with a particular focus on the TCN and Stacking models. In subsection 4.3, we outline the objectives of this section, which are to analyze and validate the performance of the proposed models through experimental results, and we also introduce the structure and progression of the subsequent subsections. We believe these modifications will improve the clarity and comprehensibility of these sections.

2. Subsection 2.1 on "Data analysis and feature engineering" provides almost very little information about the topic. Although, it's a subsection, it should provide a more in-depth discussion on the Data analysis and feature engineering.

Response:

Thank you for your valuable feedback. In response to your comment regarding the limited content in Subsection 2.1 on "Data analysis and feature engineering," we have expanded this section. We have highlighted the importance of data quality in time series forecasting and further discussed how effective data analysis and feature engineering are crucial in enhancing prediction accuracy and model efficiency. Additionally, we have included a discussion on how feature selection helps mitigate the risk of overfitting while also enhancing the model's training speed. These revisions provide a more detailed explanation of the significance of data analysis and feature engineering, and we believe they enhance the depth and clarity of this section.

3. Inclusion of a literature review is needed. The research has conducted a comparison study, however, without a proper literature review among other works, it's very hard to judge the emphasize of the results.

Response:

We have added a literature review to the paper, which will help readers gain a better understanding of our research contributions. Specifically, we have included the relevant literature review in the Introduction section, and we believe this addition will further enhance the quality of the paper.

4. A proper section dedicated to further future works will enhance the discussion presented within the paper.

Response:

Thank you for your valuable feedback. In response to your suggestion to add a dedicated section on future research directions, we have renamed the "Conclusion" section to "Conclusion and Future Research Directions" and included additional discussions on potential future research. Specifically, we emphasize that enhancing the model's ability to process large-scale datasets will be a key challenge in the context of smart grids. Moreover, we explore ways to improve the model’s ability to integrate multi-dimensional time series data, particularly with regard to factors such as weather, energy storage, and grid capacity, to improve the model's adaptability. Furthermore, we highlight the importance of broadening the model's prediction scope to better capture long-term trends and cyclical fluctuations. We believe these additions will significantly deepen the content of the paper and provide valuable guidance for future research.

Thank you for taking the time to review our paper and provide detailed feedback. Your suggestions have played a crucial role in improving the structure and content of the paper. In particular, your comments regarding the lack of overviews in subsections 2.2 and 4.3, as well as the brevity of the discussion in subsection 2.1, have helped us to better refine these sections. Furthermore, your feedback on the literature review and future research directions has added depth and clarity to the paper. We sincerely appreciate your thorough review and valuable suggestions.

Response to comments by Reviewer #2:

Comments:

1. Title the Conclusion section "Conclusion and Future Research Directions" as it includes a few details about the future work as well.

Response:

Thank you for your suggestion. We have updated the title of the Conclusion section to "Conclusion and Future Research Directions" and have included a discussion on future research directions in this section.

2. Subsection 4.3 has no information about it's heading, it directly jumps into 4.3.1 and so on. This shows inconsistent flow of information. Add neccessary information about each heading instead of leaving it blank.

Response:

Thank you for your valuable feedback. In response to your comment regarding the lack of an overview in Section 4.3, we have added an introductory paragraph at the beginning of this section to provide a clearer summary of its content.

In this revision, we outline the objectives of Section 4.3, aiming to analyze and validate the performance of the proposed model through experimental results. This section first introduces the experimental setup and data preprocessing, and then compares several deep learning models, demonstrating the performance advantages of the selected ensemble model and its potential for practical application. We believe these revisions will enhance the clarity of the section and improve the coherence of the information flow.

3. Provide more information about the evaluation metrics. The information provided is not adaequte enough.

Response:

In response to your suggestion to provide more information about the evaluation metrics, we have added a detailed explanation of the R-square, MAE, and RMSE metrics, including their principles and applications in power load forecasting. We have elaborated on how these metrics reflect model fitting, prediction error, and prediction accuracy, which helps clarify the model evaluation process. The additional content can be found in the "4.2 Evaluation Metrics" section of the manuscript. We believe that these additions will enhance the readability of the paper.

4. Perhaps a related work section could improve the quality of the research paper and add more depth and weight to the research itself

Response:

Thank you for your valuable suggestion. In response to your recommendation to include a related work section, we have made corresponding additions in Section 3.1 "Step 1: Stacking Part Construction." At the beginning of this section, we introduced research on the application of the Stacking model in forecasting tasks, providing a theoretical foundation for the introduction of the BiStacking model. Additionally, at the start of Section 3.2 "Step 2: Deep Learning Part Construction," we referenced relevant studies on TCN-LSTM and TCN-GRU to offer strong background support for the combination of BiStacking and TCN-GRU. Finally, at the beginning of Section 3.3 "Framework of the Forecasting Model," we included a brief description of the advantages of combining BiStacking and TCN-GRU. We believe these modifications enhance the academic depth of the paper and provide readers with a more comprehensive theoretical background and research foundation.

---

## [Decision Letter · Decision Letter 2]

31 Jan 2025

PONE-D-24-40595R2A hybrid power load forecasting model combining ensemble learning and deep learning: BiStacking+TCN-GRUPLOS ONE

Dear Dr. Ma,

Thank you for submitting your manuscript to PLOS ONE. After careful consideration, we feel that it has merit but does not fully meet PLOS ONE’s publication criteria as it currently stands. Therefore, we invite you to submit a revised version of the manuscript that addresses the points raised during the review process.

We look forward to receiving your revised manuscript.

Kind regards,

Jinran Wu, PhD

Academic Editor

PLOS ONE

Journal Requirements:

Reviewers' comments:

Reviewer's Responses to Questions

**Comments to the Author**

1. If the authors have adequately addressed your comments raised in a previous round of review and you feel that this manuscript is now acceptable for publication, you may indicate that here to bypass the “Comments to the Author” section, enter your conflict of interest statement in the “Confidential to Editor” section, and submit your "Accept" recommendation.

Reviewer #3: All comments have been addressed

Reviewer #4: All comments have been addressed

2. Is the manuscript technically sound, and do the data support the conclusions?

Reviewer #3: Yes

Reviewer #4: Yes

3. Has the statistical analysis been performed appropriately and rigorously? 

Reviewer #3: Yes

Reviewer #4: No

4. Have the authors made all data underlying the findings in their manuscript fully available?

Reviewer #3: Yes

Reviewer #4: No

5. Is the manuscript presented in an intelligible fashion and written in standard English?

Reviewer #3: Yes

Reviewer #4: Yes

6. Review Comments to the Author

Reviewer #3: After reviewing the revised manuscript, one area for improvement is the literature review. While it has been expanded, it would be beneficial to include more recent studies (past 2–3 years) on hybrid forecasting models combining ensemble and deep learning to enhance the study’s relevance and academic depth. Specifically, incorporating references such as 10.1007/s10489-021-02473-5 and 10.1007/s13042-024-02302-4 would strengthen the discussion.

Reviewer #4: The manuscript proposes an innovative approach to short-term power load forecasting. The study introduces a hybrid model that integrates ensemble learning and deep learning techniques, leveraging BiStacking as an ensemble method and TCN-GRU as a deep learning framework. The research highlights the model's performance using Panama's 2020 electricity load data, demonstrating significant improvements in accuracy and efficiency through metrics such as RMSE, MAE, and R-squared. This novel model offers practical and theoretical contributions, particularly in improving grid operation stability and addressing dynamic power production conditions.

Major Comments

The manuscript needs to improve the clarity and depth of its explanations regarding the model's components and their integration. For instance, the rationale behind combining BiStacking and TCN-GRU should be elaborated, particularly how these methods complement each other in handling complex data patterns. Additionally, the manuscript should provide more details on the computational efficiency and scalability of the proposed model, especially when applied to datasets larger or more diverse than the one used in the study. Lastly, while the experimental results are comprehensive, the discussion does not sufficiently link these findings to practical applications in power grid management, which would strengthen the manuscript's contributions.

Minor Comments

The title should be streamlined for better readability; consider simplifying it while retaining key terms like "hybrid" and "forecasting." The introduction should include more recent references to contextualize the novelty of the proposed approach. Figures should include more detailed captions that explain the axes, key data points, and their relevance to the study. The abstract contains redundant phrases and could be made more concise. Certain mathematical equations lack clear variable definitions, which should be provided for better comprehension. Section 2.1 on feature engineering should offer examples or visualizations to illustrate the impact of feature selection on model performance. A more detailed explanation of the evaluation metrics, particularly the trade-offs among RMSE, MAE, and R-squared, would benefit readers unfamiliar with these terms. The authors should include a justification for their choice of Panama's dataset and discuss how the results might generalize to other regions or timeframes. The discussion section should further elaborate on the limitations of the proposed model and potential solutions. The references section needs to be checked for consistency in formatting and completeness.

7. PLOS authors have the option to publish the peer review history of their article (what does this mean? ). If published, this will include your full peer review and any attached files.

**Do you want your identity to be public for this peer review?** For information about this choice, including consent withdrawal, please see our Privacy Policy .

Reviewer #3: No

Reviewer #4: No

---

## [Author Response · Author response to Decision Letter 2]

6 Mar 2025

Dear Editors and Reviewers:

We sincerely thank the Editors and reviewers for their insightful and constructive comments on our manuscript, A Hybrid Power Load Forecasting Model using BiStacking and TCN-GRU We deeply appreciate the time and effort you dedicated to reviewing our work. Your valuable feedback has been instrumental in identifying areas for improvement and has provided essential guidance in refining the paper. Below is our detailed response to each of the reviewers' comments.

Response to comments by Reviewer #3:

Thank you for reviewing our manuscript once again and providing detailed and constructive feedback. Your suggestions are crucial for further enhancing the academic depth of our paper. We especially appreciate your insights on improving the literature review and your recommendations of key references, which have helped refine our discussion on hybrid prediction models.

Comments:

1. After reviewing the revised manuscript, one area for improvement is the literature review. While it has been expanded, it would be beneficial to include more recent studies (past 2–3 years) on hybrid forecasting models combining ensemble and deep learning to enhance the study’s relevance and academic depth. Specifically, incorporating references such as 10.1007/s10489-021-02473-5 and 10.1007/s13042-024-02302-4 would strengthen the discussion.

Response:

Thank you for your valuable feedback. We have incorporated and cited the two references you mentioned (10.1007/s10489-021-02473-5 and 10.1007/s13042-024-02302-4) in our manuscript. The innovative ideas presented in these papers are highly relevant to our research and provide significant insights for future research directions. The corresponding content has been marked in the Introduction section of the manuscript.

Response to comments by Reviewer #4:

Thank you for taking the time to review our manuscript and for providing profound and constructive feedback. Your insights have played a crucial role in enhancing the clarity, scientific rigor, and practical applicability of our paper. We are especially grateful for your suggestions regarding the integration of model components, computational efficiency, scalability, and the connection between experimental results and real-world power grid management. Furthermore, your recommendations on title optimization, literature updates, figure details, mathematical expressions, feature engineering visualization, evaluation metric explanations, and reference standardization have significantly helped us refine the paper’s presentation and academic rigor. Based on your feedback, we have carefully revised the manuscript and made detailed additions in relevant sections to further improve its quality and readability.

Comments:

1. The manuscript needs to improve the clarity and depth of its explanations regarding the model's components and their integration. For instance, the rationale behind combining BiStacking and TCN-GRU should be elaborated, particularly how these methods complement each other in handling complex data patterns.

Response:

BiStacking, as an ensemble learning method, focuses on mathematical modeling and optimization. By constructing multiple base learners and integrating their predictions, this method constructs multiple base learners and integrates their predictions to fully leverage the advantages of different base learners. Its core advantages lie in flexibility and efficiency, enabling different types of base learners to work collaboratively, thereby improving prediction accuracy. Additionally, BiStacking demonstrates strong generalization ability, making it suitable for various datasets while reducing the risk of overfitting associated with single models. Therefore, it is particularly effective in handling complex data. However, despite its advantages in dealing with complex datasets, BiStacking has limitations in capturing the long-term dependencies in time-series data.

TCN-GRU combines convolutional features with a gating mechanism to effectively capture long-term dependencies in time-series data. TCN extracts key features at different time scales through convolution operations, while GRU leverages its gating mechanism to effectively capture complex dynamics and nonlinear relationships in time-series data. TCN efficiently extracts global temporal features, whereas GRU further optimizes the transmission and retention of information. Therefore, in load forecasting tasks, TCN-GRU can effectively handle the dynamic characteristics of time-series data. In contrast, when using BiStacking alone, it may struggle to fully capture the long-term dependencies and nonlinear features inherent in time-series data.

When these two approaches are combined, BiStacking enhances the recognition and analysis of complex patterns by integrating multiple models, adjusting base model weights, optimizing the decision-making process, and improving the stability and interpretability of predictions. At the same time, TCN-GRU overcomes the limitations of single deep learning models by quickly responding to data fluctuations and enhancing the ability to process complex time-series data, thereby compensating for BiStacking’s weaknesses in temporal dependency modeling. The synergy between these two methods ultimately improves the accuracy and robustness of load forecasting.

The relevant content has been added to Section 4.3.4 BiStacking+TCN-GRU Experimental Results in the conclusion section of the paper.

2. The manuscript should provide more details on the computational efficiency and scalability of the proposed model, especially when applied to datasets larger or more diverse than the one used in the study.

Response:

Regarding computational efficiency and model scalability, the proposed BiStacking+TCN-GRU model has demonstrated strong computational efficiency in our experiments (see Section 4.3.3: Deep Learning Model Selection). By comparing the training time and prediction accuracy of state-of-the-art deep learning models under the same dataset, and comprehensively considering prediction time, accuracy, and memory consumption, the proposed BiStacking+TCN-GRU model effectively reduces computational costs while significantly improving prediction accuracy.

In terms of model scalability, the proposed BiStacking, as an ensemble learning method, can flexibly adapt to large-scale data by increasing the number of base learners or adjusting their types while optimizing based on data characteristics. To enhance the computational efficiency of deep learning models, future research could further explore incorporating an Attention mechanism into neural networks to dynamically allocate weights to temporal features, enabling more precise capture of key information. Additionally, integrating noise suppression and signal decomposition techniques in the data preprocessing stage could help uncover underlying patterns, providing stronger support for model optimization.

When applied to larger or more diverse datasets, the proposed model is well-suited for multi-feature load forecasting tasks and effectively selects relevant features, making it particularly applicable to datasets with high variability. To address the challenges posed by large-scale datasets, this study focuses on short-term load forecasting. Even with data collected hourly, the total number of samples within a year does not exceed 10,000. Within this data range, the proposed model meets forecasting requirements. To enhance the processing capability for large-scale datasets, future research could further integrate incremental learning and structural pruning techniques, along with optimization algorithms to adjust model parameters, thereby improving training efficiency on large-scale datasets.

We sincerely appreciate the reviewer’s insightful comments, which have been instrumental in refining our manuscript. The relevant additions have been incorporated and marked in the conclusion section to better highlight the contributions of this study.

3. While the experimental results are comprehensive, the discussion does not sufficiently link these findings to practical applications in power grid management, which would strengthen the manuscript's contributions.

Response:

Thank you for your valuable feedback. Your suggestions regarding the discussion of experimental results are crucial for improving the completeness of the paper and ensuring that the findings align more closely with real-world needs. Currently, the discussion lacks a strong connection between the experimental results and practical applications in power grid management. This is primarily because model deployment requires adjustments based on specific application scenarios and local conditions. Although this study does not directly involve practical applications, the experimental results were validated using real load data, ensuring the model's effectiveness in electricity load forecasting and demonstrating its potential for application in this field. Future research will further explore deployment strategies in real-world scenarios and optimize the model based on different application requirements. Once deployment conditions are clearly defined, the model will better serve the needs of power grid management, facilitating its practical implementation.

The relevant additions have been incorporated into the Conclusion and Future Research Directions section of the paper.

4. The title should be streamlined for better readability; consider simplifying it while retaining key terms like "hybrid" and "forecasting."

Response:

We sincerely appreciate the reviewer’s valuable comments. Based on your suggestion, we have revised the title from “A Hybrid Power Load Forecasting Model Combining Ensemble Learning and Deep Learning: BiStacking+TCN-GRU” to “A Hybrid Power Load Forecasting Model Using BiStacking and TCN-GRU”. We believe that this simplified title improves readability while retaining the essential keywords.

5. The introduction should include more recent references to contextualize the novelty of the proposed approach.

Response:

Thank you for your valuable suggestion. we have incorporated relevant studies from the past 2–3 years into the Introduction section to better highlight the novelty of the proposed method and to strengthen the academic background and practical relevance of the paper. The specific citations for the newly added references have been marked in the revised manuscript.

6. Figures should include more detailed captions that explain the axes, key data points, and their relevance to the study.

Response:

We sincerely appreciate the reviewer’s valuable suggestions. Based on your feedback, we have made the following modifications to the figures and charts. Specifically, we have added explanations for the x- and y-axes in the fitting plots and provided a detailed interpretation of the magnified sections in Section 4.3: Experimental Results Analysis. Additionally, for the bar chart summarizing the evaluation metrics, we have included additional explanations in the same section to clarify the results. Furthermore, we have emphasized key data points in the analysis and explored the relevance of the graphical results to the research objectives. These revisions aim to enhance readability and improve the accuracy of information transmission, ensuring that readers can better understand the figures. The specific modifications can be found in the annotated sections of the revised manuscript.

7. The abstract contains redundant phrases and could be made more concise.

Response:

Thank you for your valuable suggestion. Based on your suggestion, we have simplified the abstract by removing redundant expressions and emphasizing the core aspects of our research. We hope that this revision further enhances the clarity and readability of the paper.

8. Certain mathematical equations lack clear variable definitions, which should be provided for better comprehension.

Response:

Thank you for your valuable suggestion. The issue you raised is crucial, as providing detailed explanations of the variables in the equations helps readers better understand their meanings and roles within the formulas. We have thoroughly reviewed all equations and ensured that the definitions and descriptions of the relevant variables are complete and clear. Based on your suggestion, we have further refined the explanations of the equations to improve their comprehensibility and enhance the overall clarity of the paper.

9. Section 2.1 on feature engineering should offer examples or visualizations to illustrate the impact of feature selection on model performance.

Response:

Thank you for your valuable suggestion. To more clearly illustrate the impact of feature selection on model performance, we have added comparative charts of load forecasting results before and after feature engineering in Section 4.3.2. This visually demonstrates the improvement in prediction accuracy achieved through feature engineering, reinforcing the understanding of the importance of feature selection. Additionally, this section elaborates on the feature selection method in feature engineering, where low-correlation features are removed by calculating correlation coefficients, and experiments have been conducted to validate their impact on model prediction accuracy.

10. A more detailed explanation of the evaluation metrics, particularly the trade-offs among RMSE, MAE, and R-squared, would benefit readers unfamiliar with these terms.

Response:

Thank you for your valuable suggestion. In response to your suggestion, we have added more detailed explanations in the revised manuscript, particularly regarding the trade-offs between RMSE, MAE, and R-squared. We have clarified the definitions and characteristics of these evaluation metrics to help readers better understand their applications and significance. We believe that this revision will improve the readability of the paper, especially for readers who may not be familiar with these terms.

11. The authors should include a justification for their choice of Panama's dataset and discuss how the results might generalize to other regions or timeframes.

Response:

We sincerely appreciate the reviewer’s valuable comments. We selected the Panama power load dataset primarily because of its strong representativeness and broad applicability, making it particularly suitable for verifying the effectiveness of short-term load forecasting models. Additionally, this dataset is open-source, facilitating researchers in using and reproducing experiments, thereby promoting advancements in related research. This dataset spans a long period and exhibits significant seasonal fluctuations and complex load variation patterns, making it an ideal data source for testing the proposed hybrid forecasting model.

Regarding the generalizability of the model results, experimental findings indicate that the proposed model performs exceptionally well in power load forecasting tasks. We believe that for similar load forecasting tasks in different regions, the model can effectively adapt and generate predictions as long as historical load data from the target region is provided. For changing time periods, the manual update strategy used in the model accommodates different temporal requirements, as detailed in the Conclusion and Future Research Directions section. Additionally, the model efficiently handles multi-feature data, making it particularly suitable for complex load forecasting tasks.

Clarifying the rationale behind dataset selection provides strong support for the model’s application in multi-feature forecasting tasks. This addition not only enhances the rigor of the paper but also strengthens its practical relevance by demonstrating the model’s applicability across different regions and time periods. We sincerely appreciate the reviewer’s insightful suggestions, and the relevant additions have been incorporated into Section 4.1: Data Sources.

12. The discussion section should further elaborate on the limitations of the proposed model and potential solutions.

Response:

Thank you for your valuable suggestion. To address the limitations of the model and explore potential solutions, we have provided further elaboration:

1.Extreme Weather and Unexpected Events

Although the BiStacking+TCN-GRU model perfo

---

## [Editor Report · Decision Letter 3]

10 Mar 2025

A Hybrid Power Load Forecasting Model using BiStacking and TCN-GRU

PONE-D-24-40595R3

Dear Dr. Ma,

We’re pleased to inform you that your manuscript has been judged scientifically suitable for publication and will be formally accepted for publication once it meets all outstanding technical requirements.

Kind regards,

Jinran Wu, PhD

Academic Editor

PLOS ONE

---

## [Editor Report · Acceptance letter]

PONE-D-24-40595R3

PLOS ONE

Dear Dr. Ma,

I'm pleased to inform you that your manuscript has been deemed suitable for publication in PLOS ONE. Congratulations! Your manuscript is now being handed over to our production team.

Kind regards,

on behalf of

Dr. Jinran Wu

Academic Editor

PLOS ONE